# The conformation of the histone H3 tail inhibits association of the BPTF PHD finger with the nucleosome

Emma A Morrison[1], Samuel Bowerman[2,3], Kelli L Sylvers[1], Jeff Wereszczynski[2,3]*, Catherine A Musselman[1]*

[1]Department of Biochemistry, Carver College of Medicine, University of Iowa, Iowa City, United States; [2]Department of Physics, Illinois Institute of Technology, Chicago, Illinois; [3]Center for Molecular Study of Condensed Soft Matter, Illinois Institute of Technology, Chicago, Illinois

**Abstract** Histone tails harbor a plethora of post-translational modifications that direct the function of chromatin regulators, which recognize them through effector domains. Effector domain/histone interactions have been broadly studied, but largely using peptide fragments of histone tails. Here, we extend these studies into the nucleosome context and find that the conformation adopted by the histone H3 tails is inhibitory to BPTF PHD finger binding. Using NMR spectroscopy and MD simulations, we show that the H3 tails interact robustly but dynamically with nucleosomal DNA, substantially reducing PHD finger association. Altering the electrostatics of the H3 tail via modification or mutation increases accessibility to the PHD finger, indicating that PTM crosstalk can regulate effector domain binding by altering nucleosome conformation. Together, our results demonstrate that the nucleosome context has a dramatic impact on signaling events at the histone tails, and highlights the importance of studying histone binding in the context of the nucleosome.
DOI: https://doi.org/10.7554/eLife.31481.001

*For correspondence:
jwereszc@iit.edu (JW);
catherine-musselman@uiowa.edu (CAM)

Competing interests: The authors declare that no competing interests exist.

## Introduction

Eukaryotic DNA is packaged into the cell nucleus in the form of chromatin. This DNA/histone complex contributes to DNA compaction and restricts its accessibility, also providing a mechanism for regulating the genome. Dynamic modulation of chromatin structure is vital in all DNA-templated processes. The basic subunit of chromatin is the nucleosome, comprised of an octamer containing two copies each of histones H2A, H2B, H3, and H4, around which ~ 147 bp of DNA is wrapped. The N-termini of all four histones, as well as the C-terminus of H2A, protrude to the exterior of the nucleosome core and are commonly referred to as histone tails. These tails can be extensively post-translationally modified, which is thought to be critical in regulation of chromatin structure. Genome-wide studies have revealed that particular post-translational modifications (PTMs) are correlated with specific genomic states and/or elements, and notably, strong correlations have led to the suggestion that it is patterns of PTMs that are functionally important (*Zentner and Henikoff, 2013*). Though some histone tail PTMs have been found to directly affect chromatin array compaction (*Kan et al., 2009*; *Wang and Hayes, 2008*; *Zhou et al., 2012*; *Dhall et al., 2014*; *Mishra et al., 2016*), most are thought to act through indirect mechanisms, by recruiting regulatory complexes to modified nucleosomes or modulating their activity once there. Modified histone tails are recognized by effector domains, often referred to as histone readers. A large number of families of histone effector domains have been identified over the past two decades including bromodomains, chromodomains, and PHD fingers, and these often exist in multiples within chromatin regulators (*Musselman et al., 2012*; *Andrews et al., 2016*). A wealth of studies have examined the structural determinants of

**eLife digest** The human genome contains all the instructions needed to build the human body. However, each human cell does not read all of these instructions, which come in the form of genes encoded in the DNA. Instead, different subsets of genes are switched on in each type of cell, while the rest of the genes are switched off.

DNA within human cells is wrapped around proteins called histones, to form hundreds of thousands of structures called nucleosomes. If the DNA that encodes a gene contains a lot of nucleosomes, the DNA is not very accessible and the gene will generally be off; removing the histones or rearranging the nucleosomes can turn the gene on.

Each histone contains a region called a tail – because it protrudes like the tail of a cat – that can be chemically modified in dozens of different ways. Particular combinations of histone modifications are thought to signal how the nucleosomes should be arranged so that each gene is properly regulated. However, it is unclear how these combinations of modifications actually work because, historically, it has been difficult to study tails in the context of a nucleosome. Instead most studies had looked at tails that had been removed from the nucleosome.

Now, Morrison et al. set out to investigate how one protein, called BPTF, recognizes a specific chemical modification on the tail of a histone, referred to as H3K4me3, in the context of a human nucleosome. Unexpectedly, the experiments showed that the histone-binding domain of BPTF, which binds to H3K4me3, was impeded when the tail was attached to the nucleosome but not when it was removed from the nucleosome. Morrison et al. went on to show that this was because the histone tail is tucked onto the rest of the nucleosome and not easily accessible. Further experiments revealed that additional chemical modifications made the tail more accessible, making it easier for the histone-binding domain to bind.

Together these findings show that a combination of histone modifications acts to positively regulate the binding of a regulatory protein to H3K4me3 in the context of the nucleosome by actually regulating the nucleosome itself. The disruption of the histone signals is known to lead to a number of diseases, including cancer, autoimmune disease, and neurological disorders, and these findings could guide further research that may lead to new treatments. Yet first, much more work is needed to investigate how other histone modifications are recognized in the context of the nucleosome, and how the large number of possible combinations of histone signals affects this process.

DOI: https://doi.org/10.7554/eLife.31481.002

effector domain specificity (*Musselman et al., 2012*; *Andrews et al., 2016*). However, due to difficulties in crystalizing these domains in complex with modified nucleosomes, the mechanism of histone binding has largely been studied with peptides corresponding to segments of the histone tails.

Biochemical studies suggest that the histone tails are highly solvent accessible, and the large majority of crystal structures of the nucleosome do not resolve the tails, suggesting a high degree of conformational heterogeneity (*Böhm and Crane-Robinson, 1984*; *Rosenberg et al., 1986*; *Luger et al., 1997*). This is further supported by nuclear magnetic resonance (NMR) spectroscopy studies, in which the histone tails are found to have a high degree of conformational flexibility (*Zhou et al., 2012*; *Gao et al., 2013*). Together, this has led to a model of the nucleosome where the tails are extended into solution and fully accessible. If correct, this would suggest that effector domain binding to a nucleosome versus a histone tail peptide should be largely similar. However, there is also ample evidence that the histone tails stabilize nucleosome structure and alter DNA binding. In particular, histone tails have been shown to alter transcription factor accessibility (*Lee et al., 1993*; *Polach et al., 2000*; *Yang et al., 2005*), thermal stability of the nucleosome (*Ausio et al., 1989*; *Iwasaki et al., 2013*), and DNA wrapping/unwrapping rates (*Andresen et al., 2013*). There has also long been evidence that the tails can interact with DNA (*Cutter and Hayes, 2015*) (in some studies quite robustly), including a recent study in which it was found that the tails transiently interact with linker DNA, inhibiting the activity of histone modifying enzymes (*Stützer et al., 2016*). Notably, in the one crystal structure of the nucleosome where the tails are resolved, several of them are associated with the DNA of crystallographic symmetry mates

(*Davey et al., 2002*). In addition, in silico all-atom studies of the nucleosome have repeatedly suggested that the histone tails collapse onto core DNA (*Biswas et al., 2013*; *Li and Kono, 2016*; *Shaytan et al., 2016*; *Ikebe et al., 2016*; *Chakraborty and Loverde, 2017*), and tail dynamics have been correlated with the DNA unwinding process through free energy calculations (*Kenzaki and Takada, 2015*). Thus, our current understanding of the histone tail conformation within the nucleosome is incomplete, and we know very little about how effector domains recognize histone tails in the context of the nucleosome.

Here, we utilize NMR spectroscopy to investigate the interaction of a PHD finger with the nucleosome, specifically, the BPTF (bromodomain PHD finger transcription factor) PHD finger. The BPTF PHD finger is a well-characterized histone effector domain that recognizes the first 6 residues of the histone H3 tail, with specificity for tri-methylated lysine 4 (H3K4me3) (*Li et al., 2006*). We find that in the context of the nucleosome, the PHD finger association with the methylated H3 tail is inhibited. Using NMR and molecular dynamics (MD) simulations, we demonstrate that this inhibition is due to the conformation of the H3 tail within the nucleosome. Our data support a model where the H3 tails are collapsed onto the nucleosome core through robust interaction with DNA. However, they adopt an ensemble of heterogeneous DNA-bound conformations that are in fast exchange between one another, reconciling that they can be both DNA bound and have a high level of conformational flexibility. Furthermore, we find that modification or mutation of H3 tail residues outside the PHD finger binding region weakens tail association with DNA and increases accessibility to the PHD finger. This suggests that PTM cross-talk may be mediated by histone tail conformation within the nucleosome. Altogether, our data reveal a far more complex interface for effector domain binding as compared to histone peptides, and demonstrates that it is critical to characterize these associations in the proper context.

## Results

### The nucleosome inhibits association of the BPTF PHD finger with the H3 tail

In order to determine how the context of the nucleosome may affect the association of PHD fingers with the histone H3 tail, we utilized NMR spectroscopy to compare binding of the BPTF PHD finger to histone tail peptide and the nucleosome core particle (NCP). Specifically, we collected sequential $^1H$-$^{15}N$ heteronuclear single quantum coherence ($^1H$-$^{15}N$ HSQC) spectra on $^{15}N$-PHD upon titration of increasing concentrations of a methylated histone tail peptide or a methylated nucleosome.

Addition of a peptide corresponding to H3 residues 1–10 methylated at lysine 4 (H3(1–10)K4me3) (*Figure 1A*, left, and *Figure 1—figure supplement 1A* and *2C*) to $^{15}N$-PHD resulted in extensive chemical shift perturbations (CSPs) in resonances for residues in the binding pocket (*Figure 1C*) as previously determined by NMR and crystallographic studies (*Li et al., 2006*). The pattern of CSPs observed as a function of peptide concentration denotes low μM affinity and is consistent with previously reported affinities (*Li et al., 2006*). The methylated nucleosome was generated using a methyl lysine analogue (MLA) at histone H3 lysine 4 (H3K$_C$4me3-NCP) using human histones and the 147 base pair (bp) Widom 601 DNA sequence. Importantly, mass spec analysis of the H3K$_C$4me3 protein demonstrated that alkylation at position four was complete, with no evidence of over-alkylation, or carbamylation due to refolding in urea (*Figure 1—figure supplement 3A,B*). Addition of H3K$_C$4me3-NCP to $^{15}N$-PHD led to CSPs indicating binding (*Figure 1A*, center). Compared to the peptide, the same set of resonances is perturbed, and the majority of resonances track along the same trajectory (*Figure 1A–C*). This demonstrates that the mode of interaction between the PHD finger and H3 tail in the context of the NCP is the same as observed for free histone tail peptide and that the PHD finger does not make any significant contacts with other histone tails or the NCP core. The latter is further confirmed by demonstration that binding is dependent on methylation, as seen by titration of unmodified NCP into $^{15}N$-PHD, which does not result in any significant CSPs (*Figure 1—figure supplement 4*).

Surprisingly, though the binding pocket was the same, the magnitude of CSPs upon titration of H3K$_C$4me3-NCP was significantly smaller as compared to an equivalent addition of histone peptide (*Figure 1A,C*), and the titration with NCP does not reach saturation (i.e. a fully bound state) at the same molar ratios as does the peptide (*Figure 1—figure supplements 1C* and *2C*). If the mode of

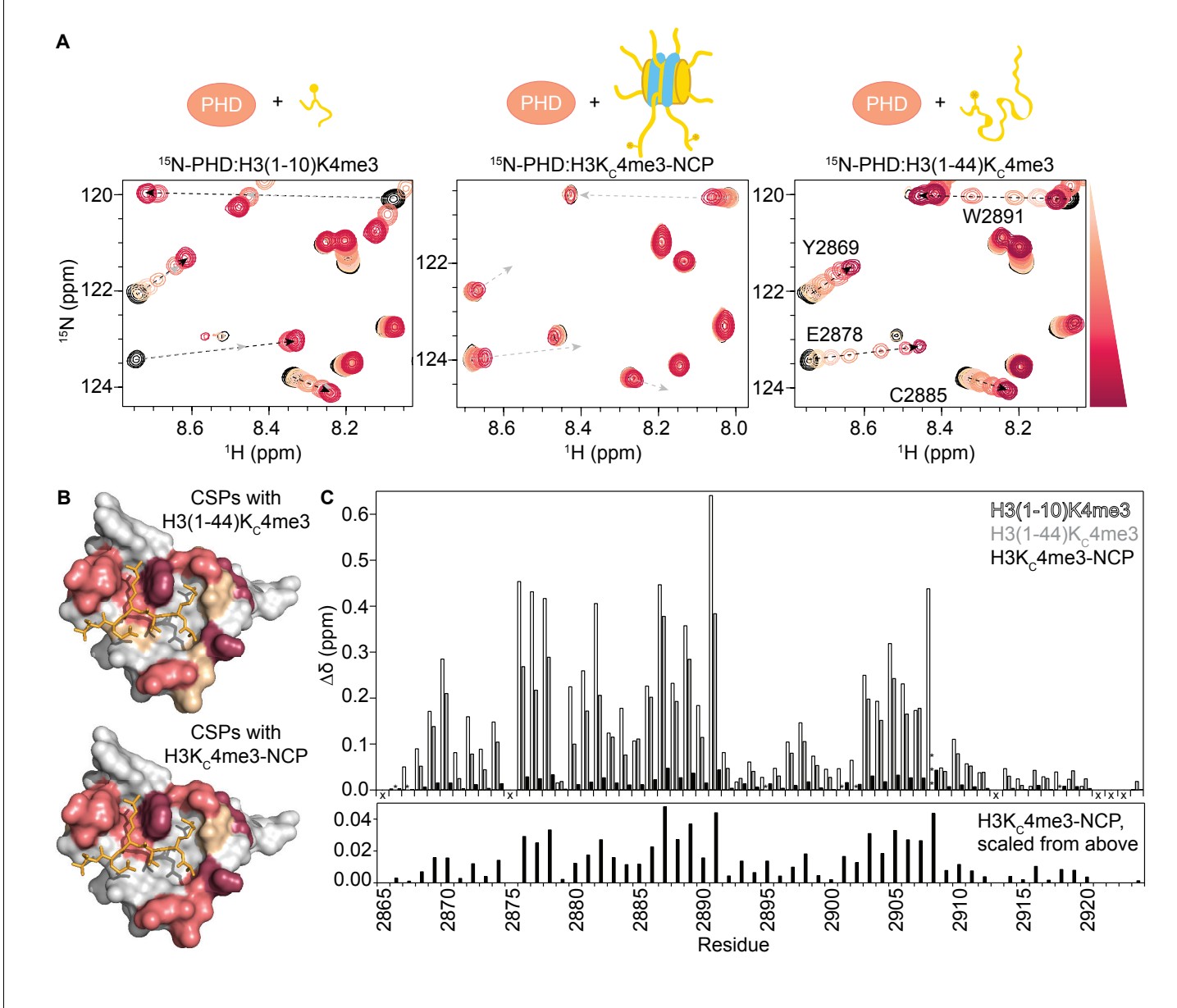

**Figure 1.** Binding of the BPTF PHD finger to H3K4me3 is inhibited in the context of the nucleosome. (**A**) Overlay of a region of the $^{1}$H-$^{15}$N HSQC (or TROSY-HSQC) spectra of 50 μM $^{15}$N-BPTF PHD upon titration of H3(1–10)K4me3 (left), H3K$_{C}$4me3-NCP (center), or H3(1–44)K$_{C}$4me3 (right). Spectra are color-coded according to ligand concentration, with apo spectra in black and shades of salmon for increasing concentrations of ligand. Spectra were collected at molar ratios (PHD:total H3K$_{C}$4me3 mark) of: 1:0, 1:0.1, 1:0.25, 1:0.5, 1:1, 1:2, and 1:5 for H3(1–10)K4me3, 1:0, 1:0.2, 1:0.5, 1:1, 1:2, 1:4 and 1:6 for H3K$_{C}$4me3-NCP, and 1:0, 1:0.1, 1:0.25, 1:0.5, 1:1, 1:2, 1:5 and 1:10 for H3(1–44)K$_{C}$4me3. Dashed arrows track the trajectory between the apo and bound states for the two peptide titrations (in black). The H3(1–44)K$_{C}$4me3 trajectories are shown (in grey) to compare titration trajectories. To account for the TROSY effect, the displayed region for the NCP titration is shifted by $J_{NH}$/2 Hz. Data was collected at 37°C in 150 mM KCl. (**B**) Surface representation of the structure of H3K4me3-bound BPTF PHD finger (PDB ID 2F6J) with residues that are significantly perturbed upon binding to H3(1–44)K$_{C}$4me3 or H3K$_{C}$4me3-NCP colored in shades of salmon (average +2, +1, +1/2 standard deviations colored in raspberry, deep salmon, and wheat, respectively). The H3 tail peptide is colored pale gold and shown in stick representation. (**C**) CSPs (Δδ) corresponding to (**A**) are plotted as a function of residue for H3(1–10)K4me3 (open bars), H3K$_{C}$4me3-NCP (filled black bars), or H3(1–44)K$_{C}$4me3 (grey bars) for the highest ligand concentration reached in each of the titrations. As expected, the largest CSPs were observed for residues within and neighboring the aromatic cage (Y2876-E2878, F2881, Y2882, and W2891) where the K4me3 group binds, and the binding pockets for H3A1 (D2908) and H3R2 (R2887-Q2889). Note that the titration with H3K$_{C}$4me3-NCP did not reach saturation. The * indicates a residue that broadens beyond detection in the fully bound state. The *** indicates a residue that is significantly perturbed along the course of the titration but broadens beyond detection in the fully bound state. An 'X' indicates that a residue is either not observable or not assigned. The Δδ for the H3K$_{C}$4me3-NCP titration is shown re-scaled below for ease of visualization.

DOI: https://doi.org/10.7554/eLife.31481.003

*Figure 1 continued on next page*

*Figure 1 continued*

The following figure supplements are available for figure 1:

**Figure supplement 1.** Full spectra of the H3 tail peptide and NCP titrations into $^{15}$N-BPTF PHD.

DOI: https://doi.org/10.7554/eLife.31481.004

**Figure supplement 2.** Further characterization of BPTF PHD finger binding to H3 tail peptides and NCP.

DOI: https://doi.org/10.7554/eLife.31481.005

**Figure supplement 3.** ESI data for histone constructs.

DOI: https://doi.org/10.7554/eLife.31481.006

**Figure supplement 4.** BPTF PHD finger does not bind to NCP in the absence of H3K$_C$4me3.

DOI: https://doi.org/10.7554/eLife.31481.007

**Figure supplement 5.** BLI data to measure the interaction between BPTF PHD finger and H3 tail in the context of peptide or NCP.

DOI: https://doi.org/10.7554/eLife.31481.008

interaction between the PHD finger and NCP is identical to the PHD-H3(1–10)K4me3 complex, as is suggested by the equivalent CSP trajectories, then the chemical shift values of the fully bound state should be identical. Under this assumption, the small shift towards the bound state seen for the NCP indicates that binding in the context of the NCP is dramatically weaker.

In general, the MLAs are robust mimetics of methylated lysines, but in some cases, they bind substantially weaker (*Seeliger et al., 2012*). A recent study extensively compared the interaction of the BPTF PHD finger with a histone peptide containing a true methylated lysine versus one containing the MLA. Though a crystal structure revealed the mechanism of association is largely the same, the authors reported a loss in binding affinity of about an order of magnitude for the MLA (*Chen et al., 2018*). Thus, we first sought to ensure that the weaker binding observed for the H3K$_C$4me3-NCP was not simply due to the analogue itself. To do this, we produced an MLA version of the histone tail alone, H3(1–44)K$_C$4me3 (which we will refer to as H3K$_C$4me3-Tail). Titration of the analogue-containing H3K$_C$4me3-Tail into $^{15}$N-PHD resulted in large CSPs (*Figure 1A*, right), immediately suggesting that the analogue is not the sole cause of the minor CSPs observed for H3K$_C$4me3-NCP. Consistent with Chen et al., binding to the MLA-containing peptide was moderately weaker as denoted by the pattern of CSPs for a subset of residues changing from intermediate (seen with the true methylated lysine) to fast exchange on the NMR time-scale (*Figure 1A*, and *Figure 1—figure supplement 1B*). Importantly, the residues perturbed are identical between the two peptides (*Figure 1B,C*), though comparison of spectra of the PHD finger saturated with either H3(1–10) K4me3 or H3K$_C$4me3-Tail reveals that the bound states are not quite identical (*Figure 1A*, and *Figure 1—figure supplement 2A*). Though for the majority of residues the resonances progress along the same trajectory from apo to bound, the difference in chemical shift (Δδ) between the free and fully bound states of the PHD is generally greater for H3(1–10)K4me3 as compared to H3K$_C$4me3-Tail (*Figure 1A,C*, and *Figure 1—figure supplement 2B*).

Fitting CSPs as a function of ligand concentration indicates that the PHD finger binds H3K$_C$4me3-Tail with a low micromolar K$_d$ of 12 ± 1 μM (*Figure 1—figure supplement 2C*, center). As saturation is not reached during the NMR titration with the H3K$_C$4me3-NCP, a robust binding affinity cannot be determined. However, the CSPs as a function of H3K$_C$4me3 mark suggest a dissociation constant in the low millimolar range (*Figure 1—figure supplement 2C*, right). Thus, the NMR data indicate that binding to the NCP is substantially weaker than binding to the free histone tail peptide. To confirm this, we utilized biolayer interferometry (BLI). A biotin-PHD construct was immobilized on streptavidin-coated sensors, and experiments were carried out with H3K$_C$4me3-Tail or H3K$_C$4me3-NCP under similar solution conditions as the NMR experiments. Clear association and dissociation curves were observed for binding to the H3K$_C$4me3-Tail with strong response signal (*Figure 1—figure supplement 5A*). Fitting the equilibrium response at the end of the association phase, the binding affinity was determined to be 7.0 ± 0.1 μM (*Figure 1—figure supplement 5B*). This is slightly higher affinity than that determined via NMR, likely due to the fact that the PHD finger is immobilized for BLI. In contrast, no association was detected with the H3K$_C$4me3-NCP at the highest concentrations used for the H3K$_C$4me3-Tail, even though it is expected to elicit a larger BLI response signal due to its significantly larger size. Even at a ten-fold higher concentration, H3K$_C$4me3-NCP produces minimal response signal (*Figure 1—figure supplement 5C*).

Together, these data demonstrate that the BPTF PHD finger adopts the same bound state with the H3 tail in the context of the nucleosome as with the free histone tail peptide, making no direct interaction with the nucleosome core. However, association is substantially inhibited in the context of the nucleosome. Though moderate effects are seen from the use of the MLA versus true methylated lysine, these do not account for the dramatically weaker association observed for the NCP. Thus, it is possible that the conformation of the nucleosome itself is causing a reduction in the observed binding affinity.

## The H3 tail experiences distinct conformations in isolation or in the native context of the NCP

To further investigate the conformation of the H3 tail (*Figure 2A*) and its potential effect on PHD finger binding, $^{15}$N-H3-NCPs were prepared by refolding octamer with $^{15}$N-labeled H3, and unlabeled H2A, H2B, and H4, and reconstituting this with the 147 bp 601 Widom DNA. Similar to previous NMR spectra on the *Drosophila* nucleosome reconstituted with 167 bp DNA and *X. laevis* nucleosome reconstituted with 187 bp DNA (*Zhou et al., 2012*; *Stützer et al., 2016*), the $^{1}$H-$^{15}$N HSQC spectrum of the $^{15}$N-H3-NCP shows only 32 of the possible 129 non-proline peaks for H3. Assignments were performed using traditional triple resonance experiments and confirm that these correspond to residues 3–36 of the N-terminal tail. The majority of these resonances have $^{1}$H chemical shifts in the range of 8.0–8.5 ppm (*Figure 2B*, left), consistent with an unstructured region, and chemical shift analysis using CSI3.0 (*Hafsa et al., 2015*) (*Figure 2—figure supplement 1A*) confirms a random coil conformation, consistent with previous studies (*Zhou et al., 2012*; *Stützer et al., 2016*). The single set of peaks observed implies that the two H3 tails within the NCP are identical. Due to the large size (~200 kDa) and resultant slow tumbling of the NCP, it is expected that the residues in the core would not be observable using this isotope labeling scheme. Thus, the observation of the H3 tail suggests that this region experiences greater intrinsic conformational dynamics than do the residues in the core. This is consistent with previous structural and biochemical studies that suggest that the histone tails are largely solvent exposed.

The ability to detect NMR resonances for the H3 tail within the nucleosome has been interpreted to reflect a disordered and predominantly accessible conformation (*Zhou et al., 2012*; *Gao et al., 2013*). However, a recent NMR study on the *X. laevis* nucleosome reconstituted with 187 bp DNA revealed differences in the nucleosomal H3 tail as compared to a histone tail peptide, and ultimately concluded that this was due to transient interactions with the linker DNA (*Stützer et al., 2016*). Here, there is no linker DNA present, thus we sought to investigate the conformation of the H3 tail in the context of the minimal NCP. Consistent with a solvent exposed conformation, changes in solution conditions, specifically temperature, KCl, and MgCl$_2$, result in global perturbations in the $^{1}$H-$^{15}$N HSQC spectra (*Figure 2—figure supplement 1B–G*), though notably, none of the conditions tested structure the H3 tail. However, similar to studies with the 187bp-nucleosome (*Stützer et al., 2016*), comparison of the $^{1}$H-$^{15}$N HMQC/HSQC spectra for an isolated peptide corresponding to residues 1–44 of the H3 tail ($^{15}$N-H3-Tail) and for the $^{15}$N-H3-NCP reveals significant chemical shift differences between the free peptide and NCP states along the full length of the H3 tail (*Figure 2B* right, 2C).

Together these data are consistent with nucleosomal H3 tails that are unstructured and highly mobile. However, chemical shift data indicate that the entire H3 tail (not just residues near the core) has a distinct conformation in the context of the NCP, consistent with an interaction within the NCP itself.

## The H3 tail robustly interacts with the nucleosome core

To determine if the H3 tail can interact with the NCP under physiological conditions, we titrated NCPs into the isolated tail (*Figure 3A*, and *Figure 3—figure supplement 1A*). Upon an initial addition of a 0.2:1 molar ratio of unlabeled NCP into $^{15}$N-H3-Tail, the sample severely aggregated, and signal was lost within the $^{1}$H-$^{15}$N HSQC spectrum. However, subsequent addition of NCP to a ratio of 0.5:1 led to a partial re-appearance of signal, with significant CSPs as compared to apo. Further addition of NCP resulted in an increase in peak intensity without perturbation in chemical shift (*Figure 3—figure supplement 1C,D*). This pattern is characteristic of a slow exchange process on the NMR timescale and indicates a robust association. Note that for slow exchange, generally a

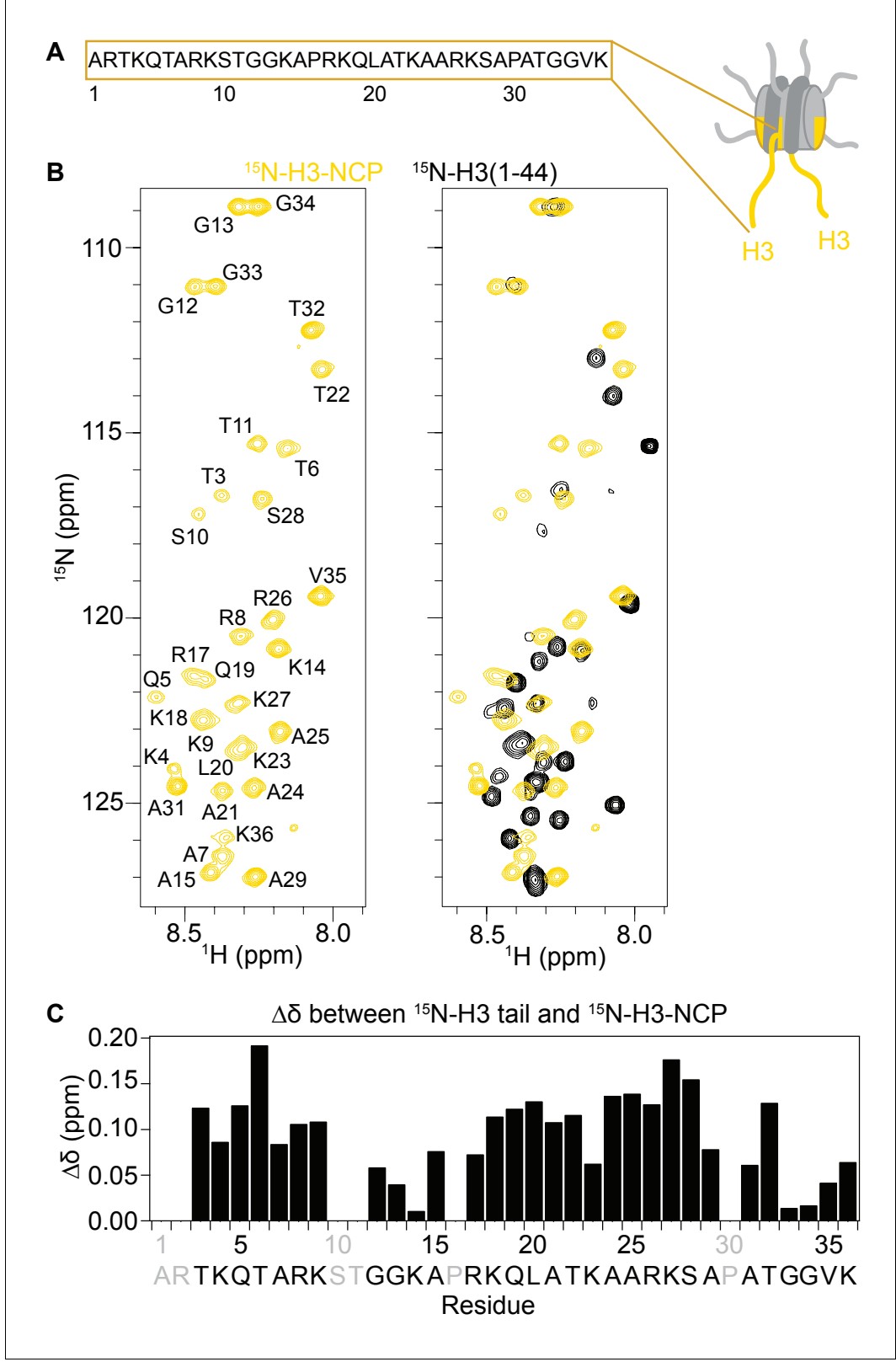

**Figure 2.** The H3 tail adopts distinct conformation(s) in the context of the nucleosome. (**A**) Cartoon depiction of the NCP with extended histone tails. H3 is colored gold, and the sequence for the H3 tail is displayed. (**B**)$^1$H-$^{15}$N HSQC or SOFAST HMQC spectra of $^{15}$N-H3-NCP (in gold) shown on the left and overlaid with $^{15}$N-H3(1–44) (black) shown on the right. Spectra were collected at 25°C in 150 mM KCl. (**C**) Chemical shift differences (Δδ) between the

*Figure 2 continued on next page*

*Figure 2 continued*

H3 tail in the context of the NCP and the isolated peptide, shown as a function of H3 tail residue. Residues labeled in grey are not observable in the $^1$H-$^{15}$N HSQC/HMQC, either because they correspond to proline or are otherwise unobservable. Though differences in chemical shift are expected for residues at the C-terminus of the tail just adjacent to the core, significant chemical shift differences are seen for the vast majority of observable residues along the entire length of the tail. Chemical shift values for both the peptide and the NCP are consistent with an intrinsically disordered conformation, suggesting that the change is not one of secondary structure. Notably, differences are largest around tandem arginine-lysine as well as serine and threonine residues.
DOI: https://doi.org/10.7554/eLife.31481.009

The following figure supplement is available for figure 2:

**Figure supplement 1.** Additional characterization of the H3 tail in the context of the NCP and peptide.
DOI: https://doi.org/10.7554/eLife.31481.010

decrease in the intensity of the apo state is observed in conjunction with an increase in intensity for the bound state. However, due to the initial aggregation, only the bound species is observed. The origin of the aggregated or phase-separated state is uncertain at this point.

This titration was repeated using tailless NCPs (tlNCPs) generated through treatment of reconstituted NCPs with trypsin (*Figure 3B*, and *Figure 3—figure supplements 1B* and *2*). Upon addition of tlNCP into $^{15}$N-H3-Tail, the same initial loss of signal was observed as for the NCP. Subsequent addition of tlNCPs led to re-appearance of signals with identical chemical shifts to those observed upon binding NCP (containing tails). This reveals that the tlNCP-bound state of H3-Tail is identical to the NCP-bound state (*Figure 3A* vs. B, and *Figure 3—figure supplement 1A–C*), and indicates that the H3-Tail-NCP interaction is not mediated through tail-tail interactions but instead through components within the core. Plotting the intensity of H3 tail bound-state resonances as a function of NCP/tlNCP concentration shows greater intensity in binding to tlNCP than to NCP throughout the titration. In addition, peak intensities plateau (indicating saturation) with tlNCP but not NCP over the concentration range tested (*Figure 3—figure supplement 1C,D*). Together, this indicates that the H3-Tail binds tlNCP tighter than NCP. This is consistent with a mechanism in which free H3 tail peptide must compete with tethered tails to associate with the core, providing additional evidence that nucleosomal H3 tails are associating with the NCP core. Plotting the CSPs as a function of residue for association with the NCP/tlNCP reveals that residues along the entire H3 tail are affected by binding (*Figure 3C*).

The structure of the isolated H3 tail peptide, the H3 tail peptide bound to the NCP *in-trans*, and the native H3 tail in the context of the NCP can be compared through overlay of the corresponding $^1$H-$^{15}$N HSQC/HMQC spectra (*Figure 3A,B*). As noted previously, there are large differences in the spectra of H3-Tail and H3-NCP (*Figure 3A*, compare black vs. gold). In contrast, the spectra for H3-Tail bound *in-trans* to the NCP or tlNCP much more closely resembles H3-NCP (*Figure 3A,B*). In fact, the majority of resonances for H3-Tail bound *in-trans* to the NCP/tlNCP lie along a linear or nearly linear trajectory between the corresponding resonances for apo H3-Tail and H3-NCP. This suggests that the *in-trans* NCP-bound state of the H3 tail peptide closely reproduces the environment of the native H3 tail, consistent with the native tail associating with the NCP core. Differences between the *in-trans* bound and native histone tails likely arise because the H3 tail is uniquely tethered to the NCP in the native state, and when binding *in-trans*, the tail is likely to associate with regions inaccessible to the restricted tail binding *in-cis*. Indeed, when comparing the chemical shift of the H3 tail peptide bound to the NCP *in-trans* and the native H3 tail, the greatest differences are in the C-terminal half of the H3 tail (*Figure 3D*). One of the largest differences is seen for lysine 36, which is immediately adjacent to where the H3 tail connects to the core histone fold and protrudes from between the DNA gyres. Additionally, there are significant differences observed for residues 24–28 and Thr32.

Notably, these data indicate that the interaction between the tails and core is quite robust, yet the fact that the resonances are detectable indicates that the tails are also conformationally dynamic. To investigate this further, MD simulations of the complete NCP were carried out. Simulations were conducted on NCPs containing unmodified H3 tails (un-NCP). Each NCP was initiated from three different H3 tail conformations: one based on the 1KX5 crystal structure (the only crystal structure that resolves all H3 histone tail residues), another in which the H3 tails were extended linearly from

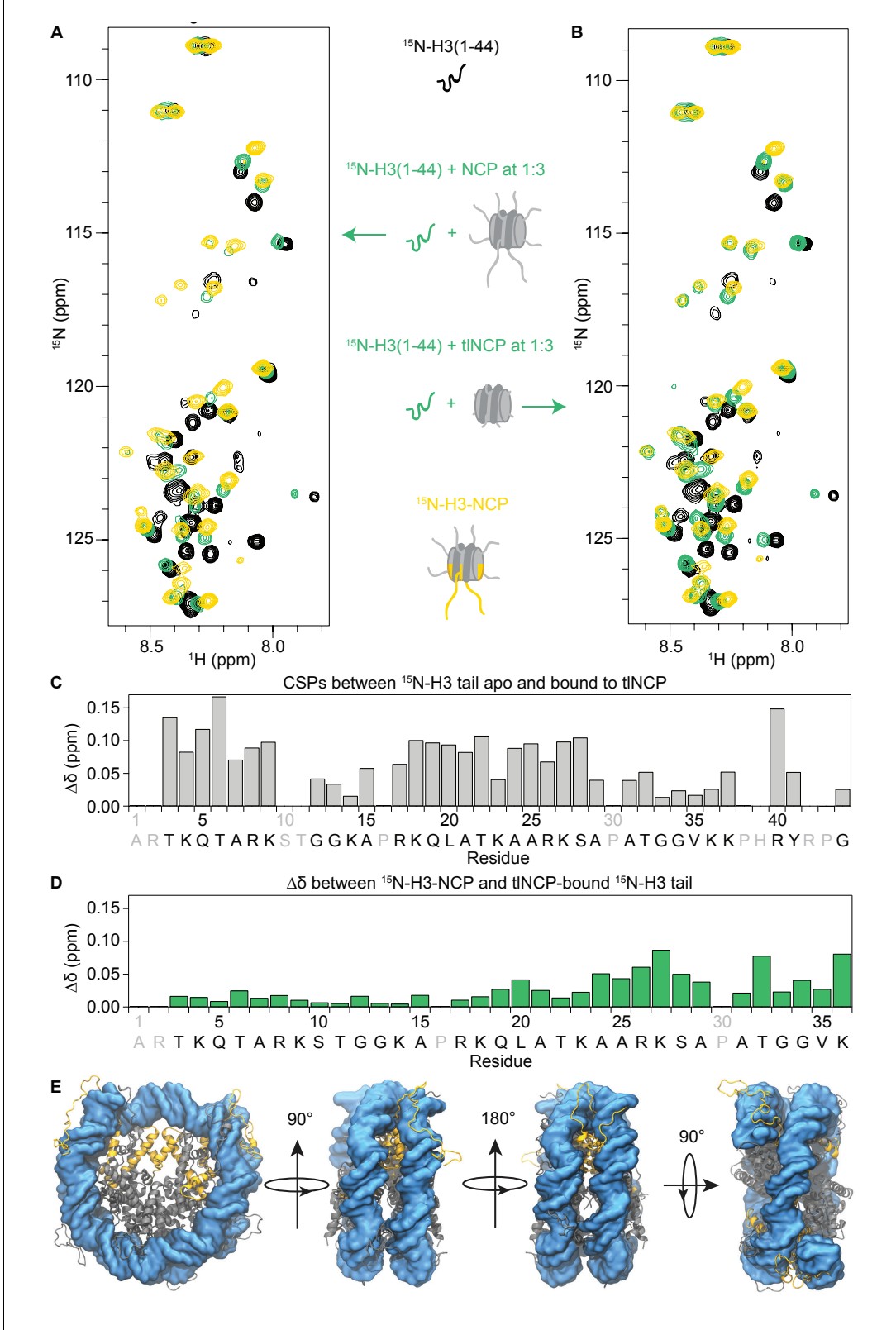

**Figure 3.** The H3 tail binds the nucleosome core. (**A**) Overlay of $^1H$-$^{15}N$ HSQC or SOFAST HMQC spectra for 70 μM $^{15}N$-H3(1–44) (black), 70 μM $^{15}N$-H3 (1–44) in the presence of unlabeled NCP at a 1:3 ratio (green), and 55 μM $^{15}N$-H3-NCP (gold). (**B**) Same as in (**A**), except that the $^{15}N$-H3(1–44) is bound to unlabeled tlNCP at a 1:3 ratio (green). These spectra indicate that H3(1–44) binds to the nucleosome core *in-trans*. (**C**) CSPs (Δδ) as a function of H3 tail residue corresponding to the difference between $^{15}N$-H3(1–44) apo and bound to tlNCP. (**D**) Chemical shift differences (Δδ) as a function of H3 tail

*Figure 3 continued on next page*

*Figure 3 continued*

residue corresponding to the difference between the native, nucleosomal H3 tail and H3(1–44) peptide bound to tlNCP. In (C) and (D), residue letters colored in grey are not observable in the $^1$H-$^{15}$N HSQC/HMQC, either because they correspond to proline or are otherwise unobservable. (E) Representative snapshot from one of the NCP simulations showing the H3 tails (gold), as well as the remaining histone tails (grey), collapsed onto the core DNA surface (blue).

DOI: https://doi.org/10.7554/eLife.31481.011

The following figure supplements are available for figure 3:

**Figure supplement 1.** Full data sets for the titration of NCP and tlNCP into $^{15}$N-H3(1–44).

DOI: https://doi.org/10.7554/eLife.31481.012

**Figure supplement 2.** Generation of tlNCPs via trypsin digest.

DOI: https://doi.org/10.7554/eLife.31481.013

**Figure supplement 3.** Simulation time course of H3 tails within unmodified NCP.

DOI: https://doi.org/10.7554/eLife.31481.014

residues 1 to 40, and a third based on *de novo* modeling of the tails. Consistent with the robust interaction between the H3 tail and the NCP core seen by NMR, simulations show that the H3 tail consistently collapses onto the NCP regardless of the initial conformation (*Figure 3E*, *Video 1*). This is measured quantitatively through calculation of the radius of gyration ($R_g$) of the full histone H3 protein, which reduced from a maximum of 45 Å to ~26 Å upon tail collapse (*Figure 3—figure supplement 3*, *Table 1*). Notably, the average root mean square deviation (RMSD) of H3 tail conformations calculated between the final frames of all simulations, after aligning NCP core conformations, is 50.0 Å, suggesting that there exists a wide range of potential conformations available to the H3 tail. If only simulations of the same initial tail conformation are considered, then this value is reduced by varying degrees to values between 30.0 Å and 41.2 Å. This reduction of 10–20 Å suggests that the final conformation of the H3 tail is influenced somewhat by the choice of initial tail structure, but that the ensemble of final states is still extensive for a single initial conformation. In addition, since the tail compacts onto the outer surface of the NCP core, a significant amount of surface area of the tails is still exposed to solvent (>2600 Å$^2$), which is in agreement with data suggesting high solvent-accessibility.

Together, these data show that the H3 tail binds robustly to the nucleosome core. The MD simulations suggest a heterogenous 'bound state' comprised of an ensemble of conformations of the H3 tail bound to the core. The fact that resonances are visible in the HSQC spectrum, and that there is only one resonance per residue, suggests that there is a fast, dynamic transition between these core-associated states.

## The H3 tail interaction with DNA mimics the chemical environment within the NCP

Several previous studies have identified an interaction between histone tails and DNA (*Cutter and Hayes, 2015*), including a recent NMR study that demonstrated interaction of the H3 tail with linker DNA in the nucleosomal context (*Stützer et al., 2016*). To test if the interaction observed here between the H3 tail and the nucleosome core is driven by DNA, we compared spectra of the H3 tail bound to a 21 bp double-stranded DNA fragment to those of the H3 tail bound to the nucleosome *in-trans* and also in its native state within the nucleosome. Comparison of the corresponding $^1$H-$^{15}$N HMQC/HSQC spectra for the DNA-bound H3-

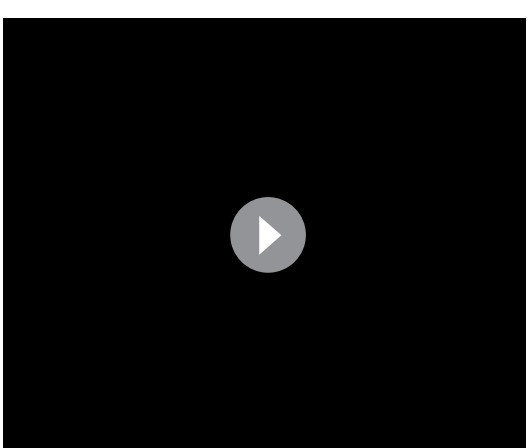

**Video 1.** End states of histone H3 tails from unmodified-NCP simulations. Protein backbone arrangement is shown in dark blue, while sidechains are represented as orange sticks. Sidechains interact with both DNA backbones (cyan and turquoise), as well as within both the major and minor grooves (grey), through a variety of conformations.

DOI: https://doi.org/10.7554/eLife.31481.016

**Table 1.** Measured quantities for the NCP simulations conducted in this study.

| Modification | Initial tails | $R_g$ (Å) | $\Delta G_{DNA\text{-}tail}$ (kcal/mol) | % Helicity | % Sheet | Solvent exposed area (Å$^2$) |
|---|---|---|---|---|---|---|
| Unmodified | 1KX5 | 24.8 ± 0.6 | −76.8 ± 6.2 | 6.1% | 2.3% | 2514 ± 61 |
| | linear | 25.1 ± 0.6 | −62.5 ± 7.1 | 6.8% | 4.0% | 2674 ± 83 |
| | MODELLER | 25.6 ± 0.6 | −73.1 ± 6.6 | 3.0% | 2.3% | 2680 ± 62 |
| | **Total** | **25.2 ± 0.3** | **−70.8 ± 4.0** | **5.3%** | **2.9%** | **2623 ± 42** |
| H3K4me3 | 1KX5 | 25.0 ± 0.5 | −82.2 ± 5.8 | 4.7% | 0.5% | 2562 ± 83 |
| | linear | 24.9 ± 0.4 | −76.3 ± 7.1 | 5.0% | 1.1% | 2697 ± 76 |
| | MODELLER | 26.1 ± 0.6 | −57.6 ± 5.3 | 7.8% | 3.3% | 2888 ± 60 |
| | **Total** | **25.3 ± 0.3** | **−72.0 ± 4.0** | **5.8%** | **1.7%** | **2716 ± 48** |
| quadAc | 1KX5 | 26.5 ± 0.5 | −54.8 ± 2.8 | 5.3% | 3.2% | 2914 ± 62 |
| | Linear | 25.5 ± 0.6 | −53.8 ± 5.9 | 5.0% | 2.9% | 3002 ± 68 |
| | MODELLER | 26.4 ± 0.4 | −47.6 ± 4.3 | 7.3% | 3.9% | 2867 ± 78 |
| | **Total** | **26.1 ± 0.3** | **−52.1 ± 2.7** | **5.9%** | **3.3%** | **2928 ± 42** |
| 4xK-Q | 1KX5 | 26.0 ± 0.4 | −59.3 ± 4.3 | 6.6% | 1.4% | 2666 ± 58 |
| | Linear | 24.6 ± 0.5 | −52.3 ± 5.1 | 6.1% | 3.7% | 2736 ± 44 |
| | MODELLER | 25.4 ± 0.5 | −57.6 ± 5.3 | 4.5% | 3.3% | 2733 ± 71 |
| | **Total** | **25.3 ± 0.3** | **−55.2 ± 2.9** | **5.7%** | **2.8%** | **2712 ± 34** |
| H3K4me3/4xK-Q | 1KX5 | 25.5 ± 0.5 | −65.4 ± 5.6 | 8.0% | 0.9% | 2744 ± 55 |
| | Linear | 25.3 ± 0.5 | −57.3 ± 5.6 | 5.9% | 3.7% | 2798 ± 88 |
| | MODELLER | 26.6 ± 0.5 | −50.3 ± 6.1 | 3.6% | 4.3% | 2784 ± 59 |
| | **Total** | **25.8 ± 0.3** | **−57.7 ± 2.7** | **5.8%** | **2.9%** | **2775 ± 40** |
| H3K4me3/3xR-A | 1KX5 | 25.3 ± 0.6 | −55.1 ± 3.9 | 7.8% | 1.9% | 2461 ± 52 |
| | Linear | 25.1 ± 0.6 | −38.9 ± 3.9 | 10.6% | 1.7% | 2567 ± 79 |
| | MODELLER | 25.7 ± 0.6 | −41.4 ± 4.2 | 8.0% | 2.4% | 2574 ± 60 |
| | **Total** | **25.4 ± 0.6** | **−45.7 ± 2.7** | **8.8%** | **2.0%** | **2534 ± 39** |

DOI: https://doi.org/10.7554/eLife.31481.015

Tail with the H3-NCP reveals marked similarities (*Figure 4—figure supplement 1*). As was seen for NCP/tlNCP-bound H3-Tail, the majority of resonances for the DNA-bound H3-Tail lie along a linear or near-linear trajectory between the chemical shift values for the apo H3-Tail and the H3-NCP (*Figure 4A*). This strongly suggests that the interaction of the H3 tail with the nucleosome core is driven through contacts with DNA. This is very similar to previous observations for the *X. laevis* 187bp-nucleosome containing linker DNA (*Stützer et al., 2016*) and reveals that, even in the absence of accessible linker DNA, the native histone H3 tails are DNA bound. Notably, the chemical shift values for the tlNCP-bound state are more similar to the native H3 tail than those for the 21bp-DNA-bound state (*Figure 4B*). This is most notable for arginine residues and, to a lesser extent, their neighboring residues (see residues 5–9, 17–18, and 24–27), which still deviate considerably in chemical shift between the DNA-bound and native H3 tails whereas binding to tlNCP much better reproduces the native NCP chemical shift. It is possible that the arginines, which fit into the minor groove of DNA, are more sensitive to DNA shape (linear vs. bent).

Consistent with the experimental data, the three-dimensional distribution of tail atoms in the MD simulation of the un-NCP shows that tails package primarily onto the core DNA surface (*Figure 5A*). MM-GBSA analysis on a per residue basis suggests that association is driven by arginines and, to a lesser extent, lysines, which is aligned with the NMR CSP data (*Figure 5C*). By calculating the sum of contributions from H3 tail residues for the Gibbs free energy (ΔG) of DNA binding, it is observed that the tails bind to DNA at a ΔG of −70.8 ± 4.0 kcal/mol. In contrast, the interaction energy between H3 tail residues and other histone components is a disfavorable 5.2 ± 0.9 kcal/mol. We note that the MM/GBSA analysis method presented here involves several approximations, including a mean-field solvent model and the lack of the solute configurational entropy change, therefore the

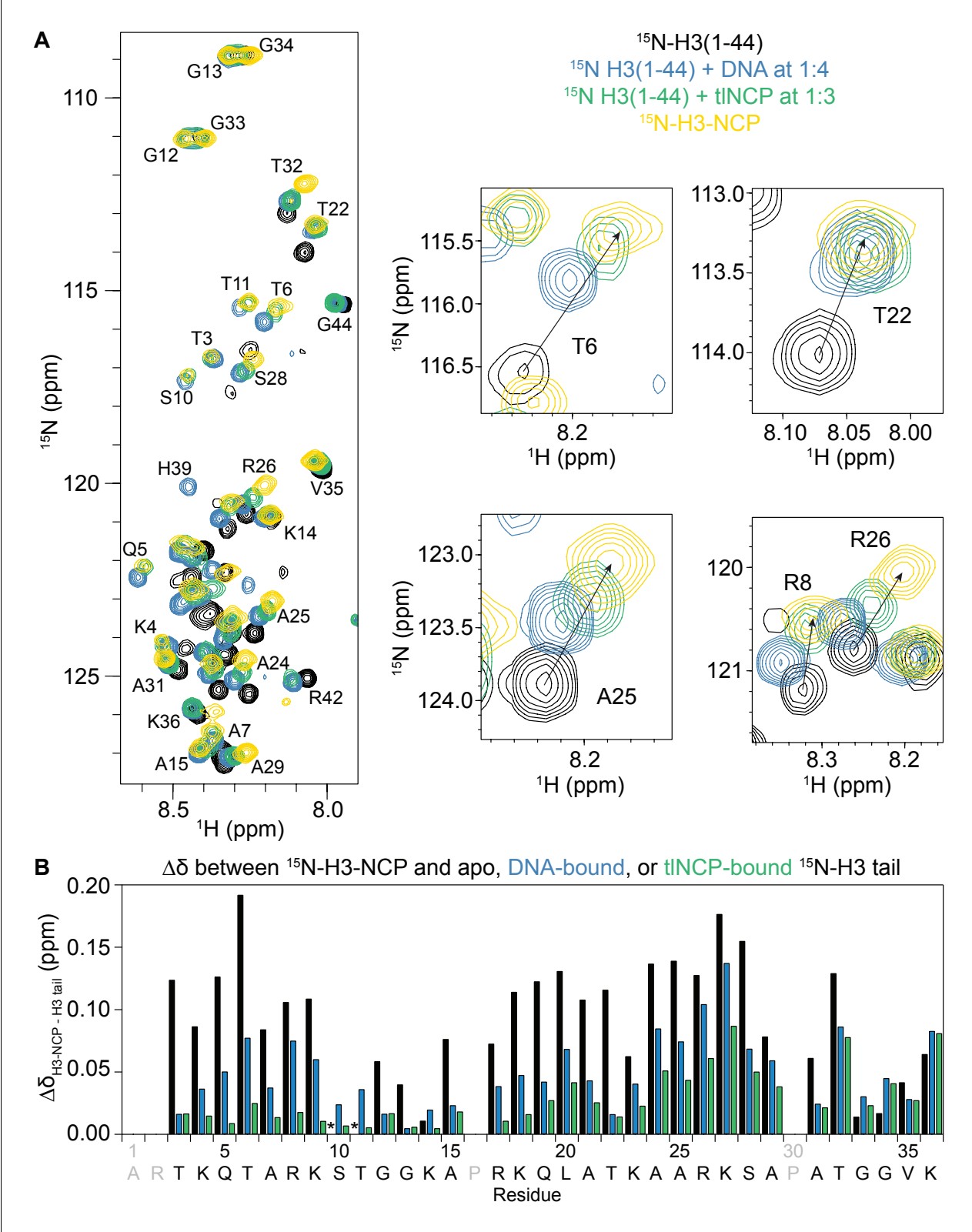

**Figure 4.** The NCP- and DNA-bound states of the H3 tail peptide mirror the H3 tail conformation in the context of the nucleosome. (**A**) Overlay of $^1$H-$^{15}$N HSQC or SOFAST HMQC spectra of $^{15}$N-H3(1–44) (black), $^{15}$N-H3(1–44) bound to DNA (blue) or tlNCP (green), and $^{15}$N-H3-NCP (gold). Expanded regions of the overlay are shown for selected residues for comparison of histone tail states. Arrows connect the apo $^{15}$N-H3(1–44) to the native $^{15}$N-H3-NCP. (**B**) Chemical shift differences (Δδ) as a function of H3 tail residue corresponding to the difference between the native, nucleosomal

*Figure 4 continued on next page*

*Figure 4 continued*

H3 and H3(1–44) either apo (black), DNA-bound (blue), or tlNCP-bound (green). Residue letters colored in grey are not observable in the $^1$H-$^{15}$N HSQC/HMQC, either because they correspond to proline or are otherwise unobservable. The * indicates that a residue is unobservable.
DOI: https://doi.org/10.7554/eLife.31481.017
The following figure supplement is available for figure 4:

**Figure supplement 1.** The H3 tail binds DNA.
DOI: https://doi.org/10.7554/eLife.31481.018

free energy change values presented should only be interpreted qualitatively (*Hou et al., 2011*; *Rastelli et al., 2010*; *Genheden et al., 2011*). Despite these limitations, our results demonstrate that interactions between the H3 tails and the NCP core are robust and favorable, and driven by preference for the DNA surface.

While some previous simulations, secondary structure predictions, and circular dichroism studies have suggested that the H3 tail adopts distinct α-helical portions (*Banères et al., 1997*; *Wang et al., 2000*; *Li and Kono, 2016*; *Ikebe et al., 2016*; *Potoyan and Papoian, 2011*), others present a tail that is largely unstructured (*Shaytan et al., 2016*; *Erler et al., 2014*; *Roccatano et al., 2007*). Here, consistent with the NMR analysis (see above), simulation results also suggest that the H3 tails are largely unstructured when bound to the nucleosome core, as determined by a DSSP characterization. This reveals that, on average, tail residues in the un-NCP are only 5.5% α-helical and 2.9% β-strand (*Table 1*). Inspecting on a per-residue basis shows that random coil is the preferred state of the entire tail (*Figure 5—figure supplement 1*). Identical calculations were also performed for NCPs containing H3K4me3 (H3K4me3-NCP). Notably, a nearly identical three-dimensional distribution is observed for tails containing H3K4me3 (*Figure 5B*, *Figure 5—figure supplement 2* and *Video 2*). The ΔG of DNA binding of the H3 tails was also not significantly altered by the presence of the tri-methyl modification (ΔG = −72.0 ± 4.0 kcal/mol), and there were no effects on secondary structure, with 6.0% α-helical and 1.6% β-strand configurations. In agreement with this, an $^1$H-$^{15}$N HSQC spectrum of the $^{15}$N-H3K$_C$4me3-NCP only demonstrates perturbations in residues around lysine 4, whereas the remainder of the tail is not perturbed, confirming that methylation of lysine four does not substantially alter the interaction of the H3 tail with the nucleosome core (*Figure 5D*).

Together, these data reveal that the H3 tail is binding the DNA component within the context of the NCP. NMR and MD simulations demonstrate that the H3 tail is robustly associated with the nucleosomal DNA. The interaction is driven largely by basic residues suggesting an electrostatic interaction. Consistent with this, spectra recorded on $^{15}$N-H3-NCP upon increasing concentrations of mono-valent ions results in CSPs that follow a trajectory roughly towards the free peptide (*Figure 2—figure supplement 1G*). Notably, methylation of lysine four does not release the H3 tail from the DNA.

## The H3 tail-DNA interaction inhibits binding by effector domains, mirroring nucleosomal conditions

To test if the interaction between the H3 tail and DNA is the cause of the decreased association between the BPTF PHD finger and H3K$_C$4me3-NCP, we probed the interaction between the PHD finger and H3 tail peptide in the presence of DNA. To this end, $^1$H-$^{15}$N HSQC spectra were collected on $^{15}$N-labeled PHD finger upon titration of unlabeled H3 tail peptide (H3(1–10)K4me3 or H3K$_C$4me3-Tail) that was pre-bound to a 21 bp DNA at a 1:2 molar ratio. The titration of DNA-bound H3(1–10)K4me3 into $^{15}$N-PHD yielded very similar CSPs as were observed upon titration of H3(1–10)K4me3 alone (*Figure 6A*). CSPs are observed along the same trajectories from apo to bound between the two titrations, indicating that the presence of DNA does not alter the binding mode. Comparison of the CSP as a function of molar ratio of ligand between histone tail alone and histone tail pre-bound to DNA reveals only a slightly lower population in the bound state when DNA is present (*Figure 6A*). In contrast, a large effect was observed for the full-length tail peptide, H3K$_C$4me3-Tail. Titration of H3K$_C$4me3-Tail pre-bound to DNA into $^{15}$N-PHD leads to CSPs that, similar to the shorter peptide, progress along the same trajectories from apo to bound as compared to H3K$_C$4me3-Tail alone. However, with the longer peptide, comparison of the CSPs as a function of

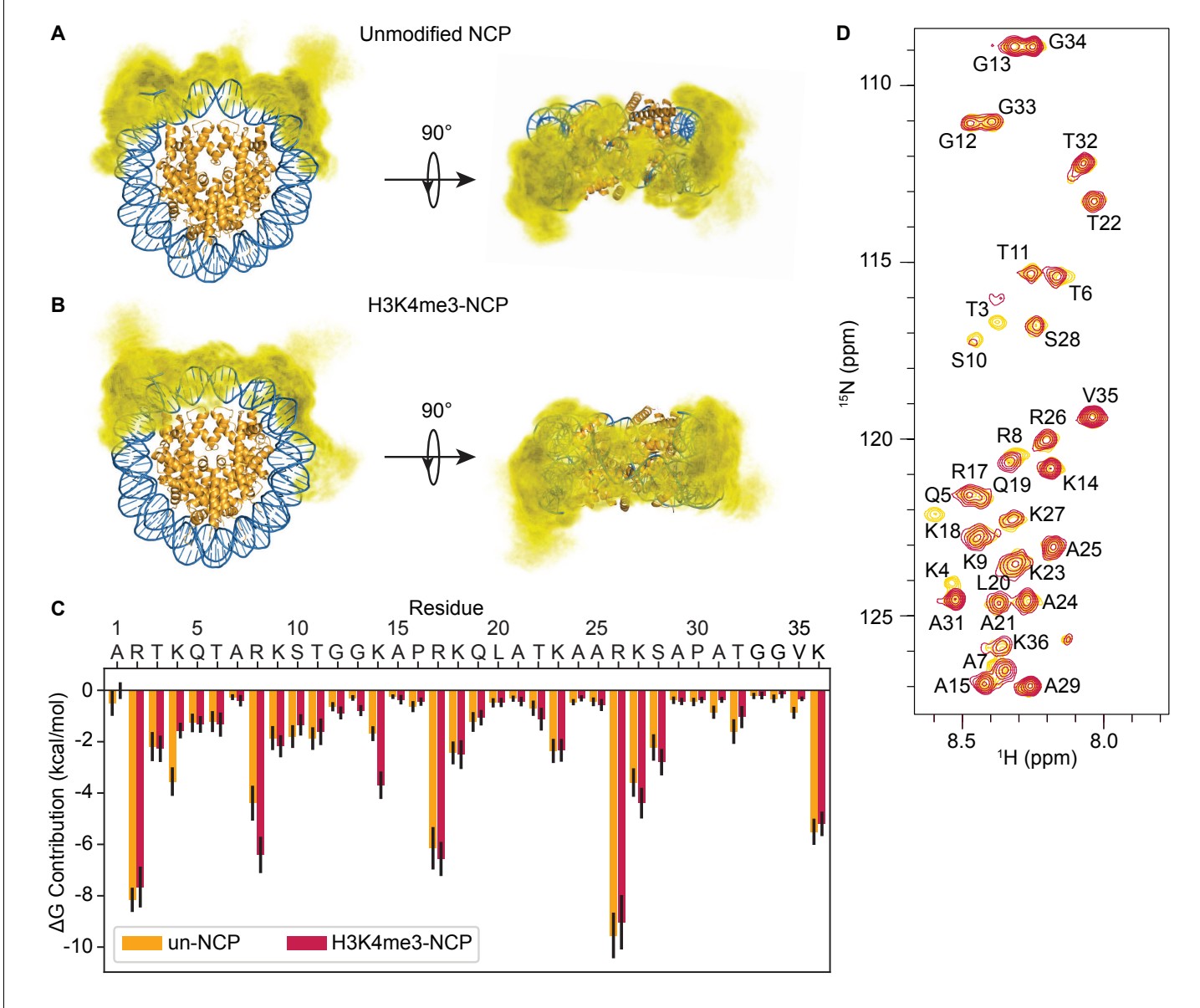

**Figure 5.** The H3 tails adopt a heterogeneous ensemble of DNA bound states. Three-dimensional probability distribution of the locations of heavy atoms within H3 N-terminal tails for the unmodified (**A**) and H3K4me3-NCP (**B**) simulations, superimposed upon the crystal structure (PDBID 1KX5). Histones are shown in gold and DNA is shown in blue. Darker regions represent areas of higher probability, and lighter regions depict areas of lower probability. (**C**) MM-GBSA analysis of the per-residue contributions to DNA binding of H3 tail residues in the unmodified (gold) and H3K4me3 (raspberry) NCP simulations. (**D**) Overlay of $^1$H-$^{15}$N HSQC spectra of $^{15}$N-H3-NCP (gold) and $^{15}$N-H3K$_C$4me3-NCP (raspberry), which indicates that the MLA installed at K4 only causes local perturbations to the H3 tail. Residues are labeled based on assignments for $^{15}$N-H3-NCP.

DOI: https://doi.org/10.7554/eLife.31481.019

The following figure supplements are available for figure 5:

**Figure supplement 1.** Secondary structure analysis of unmodified or acetylated H3 tails within the NCP.
DOI: https://doi.org/10.7554/eLife.31481.020

**Figure supplement 2.** Simulation time course of H3 tails within H3K4me3-NCP.
DOI: https://doi.org/10.7554/eLife.31481.021

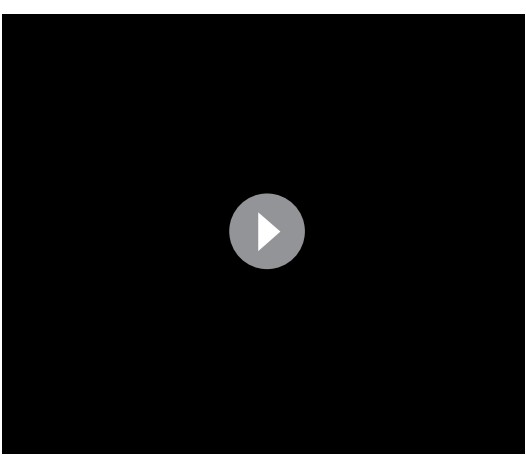

**Video 2.** End states of histone H3 tails from H3K4me3-NCP simulations. Protein backbone arrangement is shown in dark blue, while sidechains are represented as orange sticks. Sidechains interact with both DNA backbones (cyan and turquoise), as well as within both the major and minor grooves (grey), through a variety of conformations.

DOI: https://doi.org/10.7554/eLife.31481.022

ligand indicates a much smaller population of PHD in the bound state for each PHD:peptide ratio when the peptide is pre-bound to DNA than for the peptide alone (*Figure 6B*). Fitting the CSPs as a function of peptide concentration yields an apparent $K_d$ of 180 ± 30 µM for H3K$_C$4me3-Tail, which is 15-fold weaker than peptide alone (12 ± 1 µM), revealing that the interaction between the H3 tail and PHD is decreased due to competitive binding of the H3 tail to DNA (*Figure 6C*). Peak broadening suggests the possibility of a ternary complex, where the PHD finger and DNA could be simultaneously binding to different regions of H3K$_C$4me3-Tail, but this would require further investigation to confirm.

Together these data demonstrate that association of the H3 tail with DNA does indeed inhibit association of the PHD finger. A significant effect is only seen with the full length H3 tail, indicating that multiple contacts along the length of the tail are required for a high avidity interaction with DNA. Although the effect is not as strong as that observed for the H3K$_C$4me3-NCP (*Figure 1A*, center), it is expected that in the context of the nucleosome, where the tail is tethered close to the DNA, the greatly increased effective local concentration would increase the apparent affinity between the histone tail and DNA and thus its competition for PHD finger binding.

## Modification or mutation of the H3 tail weakens binding to DNA

Our results show that the H3 tail is stabilized on the nucleosomal DNA through multiple contacts along the entire length of the tail. Given this, it is possible that a variety of PTMs may alter this interaction and thus the H3 tail conformation and accessibility. Notably, our results reveal that H3K4me3 does not significantly alter the H3 tail-DNA interactions (discussed above, *Figure 5*), which is likely due to the fact that methylation of lysine does not alter the side-chain charge. However, there are a number of PTMs, such as lysine acetylation and other acylations, arginine citrullination, and serine/threonine/tyrosine phosphorylation, that alter the side-chain charge of histone residues and thus might perturb the electrostatic interaction between the histone tails and nucleosomal DNA.

Acetylation neutralizes the positive charge on the lysine side-chain. To investigate how acetylation along the H3 tail alters the tail-DNA interaction we conducted MD simulations of NCPs containing acetylation on lysines 14, 18, 23, and 27 along the H3 tail (quadAc-NCP) (*Figure 7—figure supplement 1*). Interestingly, the H3 tails in the quadAc-NCP compact to a similar extent as in the un-NCP and H3K4me3-NCP, with an $R_g$ of 26.5 ± 0.4 Å for H3 (*Table 1*). Also, similar to the un-NCP and H3K4me3-NCP the quadAc-H3 tails compact primarily onto the nucleosomal DNA (*Figure 7A*, *Video 3*). The quadAc-H3 tail shows no change in secondary structure, with an average helicity of 5.9% and an average β-strand content of 3.3% (*Figure 5—figure supplement 1*, *Table 1*). However, the interaction energies between the DNA and H3 tail are significantly weakened (ΔG = −52.1 +/- 2.7 kcal/mol, p-value of 0.0003) in the quadAc-NCP (*Figure 7B*, *Table 1*). This is in-line with previous studies using UV-induced laser-crosslinking, in which it was proposed that the histone tail-DNA interaction persists even upon acetylation, but is weakened (*Mutskov et al., 1998*). We note that the acetylated lysines only account for 33% of the total change in free energy (6.2 kcal/mol of the total 18.9 kcal/mol disfavorable shift), while the immediate neighbors of the acetylated lysines combine to contribute an additional 37% of the difference (6.9 kcal/mol). Other residues that experience significant drops in binding strength include A1 (ΔΔG = 1.7 kcal/mol, 9% of total), K4 (ΔΔG = 1.8 kcal/mol, 9% of total), and S10 (ΔΔG = 0.9 kcal/mol, 5% of total). Additionally, the average amount of solvent-exposed surface area of the H3 tails is increased in the quadAc-NCP to 2928 ± 42 Å$^2$ (*Table 1*), in

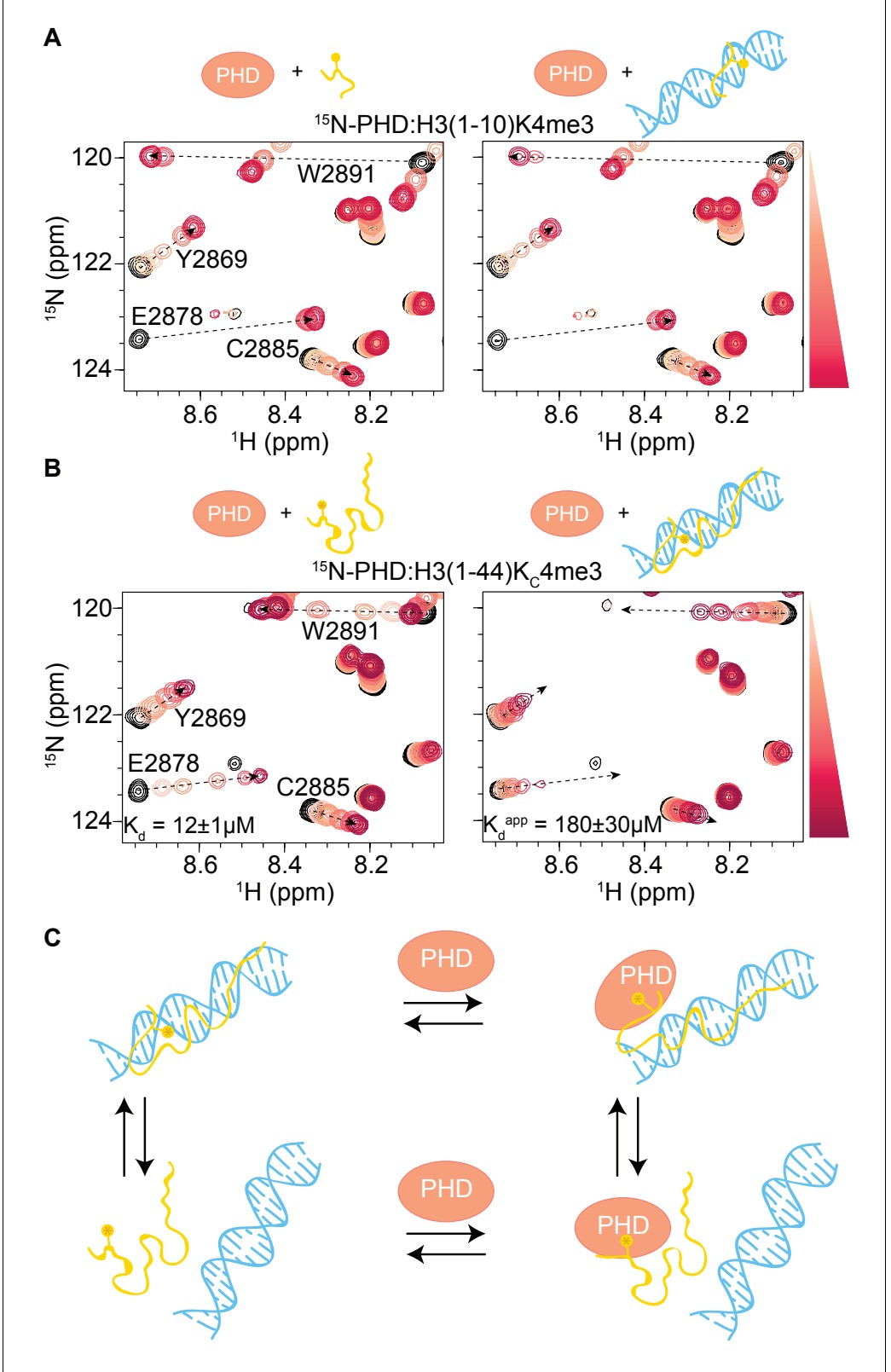

**Figure 6.** H3 tail-DNA interactions inhibit PHD finger binding. (A) Overlay of a region of the $^1$H-$^{15}$N HSQC spectra of 50 μM $^{15}$N-BPTF PHD upon titration of H3(1–10)K4me3 (left, as shown in *Figure 1A*), or H3(1–10)K4me3 pre-bound to DNA at a ratio of 1:2 (right). (B) Same as in (A) but for the H3(1–44)$K_C$4me3 peptide. Spectra are color-coded according to peptide concentration, with apo spectra in black and shades of salmon for increasing

*Figure 6 continued on next page*

*Figure 6 continued*

concentrations of peptide. Spectra are shown for molar ratios (PHD:peptide) of: 1:0, 1:0.1, 1:0.25, 1:0.5, 1:1, 1:2, and 1:5 for H3(1–10)K4me3 and 1:0, 1:0.25, 1:0.5, 1:1, 1:2, 1:5 and 1:10 for H3(1–44)$K_C$4me3. Dashed arrows track the trajectory between apo and bound states for peptide alone and are shown as visual aids for titrations where peptide was pre-bound to DNA. Binding affinities determined from the NMR data as described in Materials and methods are displayed. (C) Cartoon model to depict the linked equilibria of H3 tail peptide-DNA binding and PHD finger-H3 tail peptide binding. It is possible that the H3 tail could partially release from DNA and interact with the PHD finger, but the existence of this type of ternary complex is not conclusively shown from the data. The competitive H3 tail peptide-DNA binding is significantly greater with the longer tail peptide.
DOI: https://doi.org/10.7554/eLife.31481.023

part due to the increased size of the acetylated side chains. This increase in solvent-accessibility may provide a larger site for interaction with binding partners. Together, these results show that acetylation of lysines 14, 18, 23, and 27 reduce the strength of H3 tail interactions with DNA by removing not only direct interactions formed by these residues, but also by consequentially weakening the interactions of neighboring cationic and polar residues with the DNA.

While results show that neutralization of lysines substantially weakens tail-DNA binding, simulations also reveal that interactions between arginines and DNA provide the largest individual residue contributions. To probe the effects of arginine neutralization, we conducted simulations of modified NCPs where the arginine residues that are not implicated in PHD binding (R8, R17, and R26) were mutated to alanine in the presence of K4me3 (H3K4me3/3xR-A-NCP). As was seen with acetylation of lysine, the H3 tails in these simulations compacted to a similar extent as was seen for the unmodified NCP (H3 $R_g$ = 25.4 + /- 0.3 Å), though notably, they collapsed at a slower rate than the other NCPs (*Figure 7—figure supplement 2*). The H3K4me3/3xR-A tails occupy a similar volume of the DNA surface as compared to K4me3 alone (*Figure 7A*, right vs. *Figure 5B*, *Video 4*). There was a small increase in the average helical content (*Table 1*), however each residue was still disordered in over 50% of the simulations (*Figure 7—figure supplement 3*). In comparison to the unmodified NCP, the mutated tails have a decreased level of solvent exposed surface area (2534 ± 39 Å²), largely due to the decreased size of the alanine side-chain. Most notably, however, is that the strength of DNA binding by the mutated tail is the lowest of all NCPs tested (ΔG = −45.1 + /- 2.7 kcal/mol), and this decrease in interaction strength can be almost entirely attributed directly to the alanine mutations (*Figure 7C*), with contributions from neighboring lysines only modestly affected by the mutations. This is in stark contrast to the quadAc-NCP, which displayed a correlation between PTMs at lysine sites and a reduced interaction strength of neighboring residues with the nucleosomal DNA.

Together this reveals that mutation or modification along the H3 tail weakens, but does not abolish, the H3 tail interaction with DNA, which is mediated by a number of contacts along the tail. This suggests that modifications outside of an effector domain binding site might alter accessibility to interaction with an effector domain, mediating cross-talk between PTMs.

## Modification and mutation of the H3 tail increase accessibility to the PHD finger

To determine if these mutations and modifications increase accessibility for PHD finger binding, we used NMR. Specifically, we tested PHD finger binding to three distinct H3$K_C$4me3-NCPs: one in which R8, R17, and R26 were mutated to alanine (H3$K_C$4me3/3xR-A-NCP), one in which K14, K18, K23, and K27 were mutated to glutamine (H3$K_C$4me3/4xK-Q-NCP), and one in which S10 and S28 were phosphorylated using Aurora B kinase (H3$K_C$4me3/phos-NCP). Note that glutamine is a commonly used acetyl-lysine mimic because the polar side chain of glutamine mimics that of the neutralized lysine. To confirm the validity of this mimic, we compared the simulations of NCPs containing acetylated lysine with those containing glutamine at the same positions (with and without K4me3) and found that they were in good agreement with one another for the $R_g$ value for H3, secondary structure, and interaction energy with DNA (*Figure 7—figure supplements 4–6*, *Videos 5* and *6*). The only difference was in the solvent exposed surface area, but this is entirely accounted for by the acetylated lysine being physically larger than its glutamine counterpart (*Figure 7—figure supplement 7*).

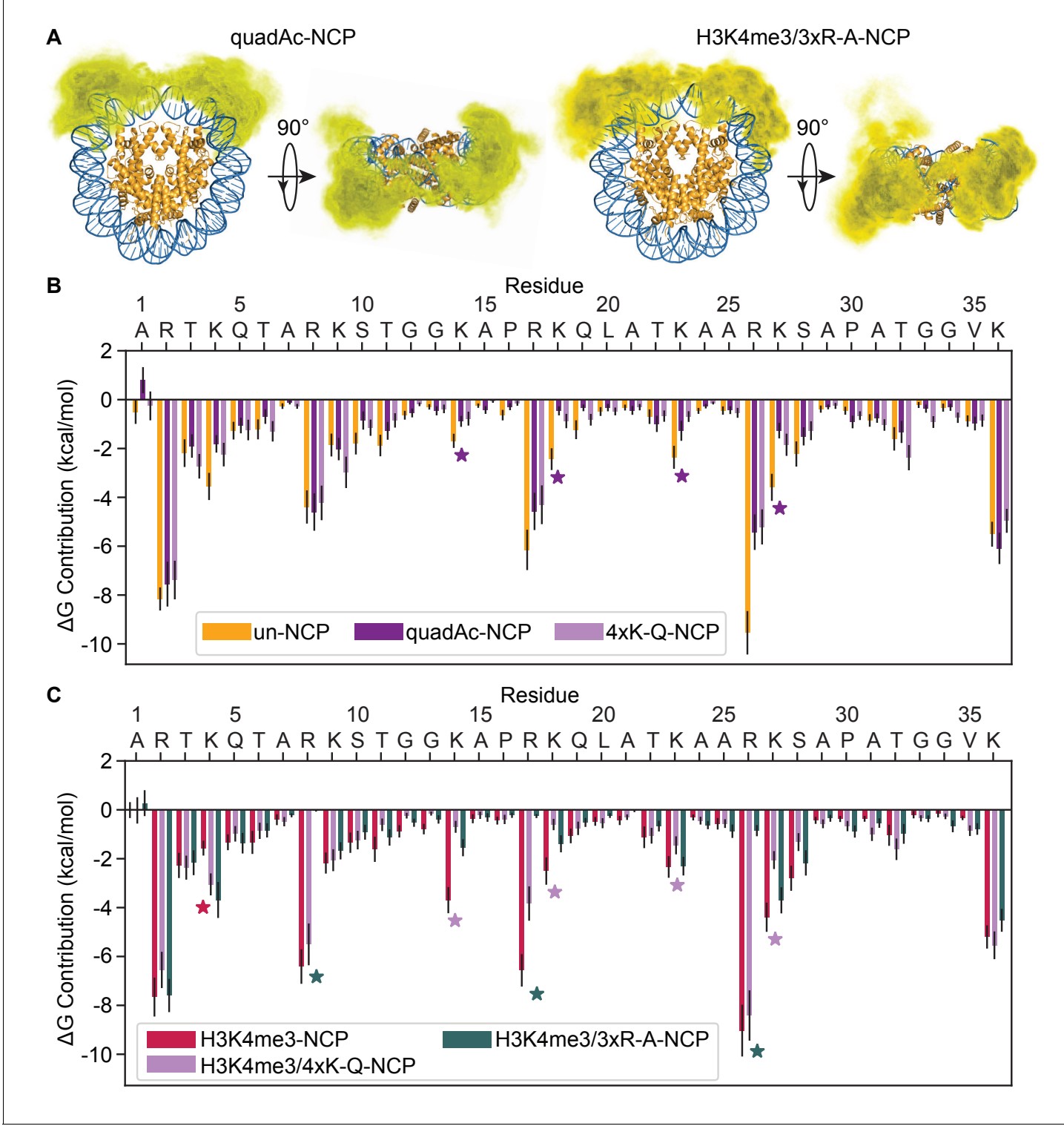

**Figure 7.** Modification and mutation of the H3 tails leads to a partial release from nucleosomal DNA. (**A**) Three-dimensional probability distribution of the locations of heavy atoms within H3 N-terminal tails for the quadAc-NCP (left) and H3K4me3/3xR-A-NCP (right) simulations, superimposed upon the crystal structure (PDBID 1KX5). Histones are shown in gold and DNA in blue. Darker regions represent areas of higher probability, and lighter regions depict areas of lower probability. (**B**) MM-GBSA analysis of the per-residue contributions to DNA binding of H3 N-terminal tail residues in the unmodified (gold), quadAc (dark purple), and 4xK-Q (lavender) NCP simulations. (**C**) MM-GBSA analysis of the per-residue contributions to DNA binding of H3 N-terminal tail residues in the H3K4me3 (raspberry), H3K4me3/4xK-Q (lavender), and H3K4me3/3xR-A (dark green) NCP simulations.

DOI: https://doi.org/10.7554/eLife.31481.024

*Figure 7 continued on next page*

*Figure 7 continued*

The following figure supplements are available for figure 7:

**Figure supplement 1.** Simulation time course of H3 tails within quadAc-NCP.
DOI: https://doi.org/10.7554/eLife.31481.025
**Figure supplement 2.** Simulation time course of H3 tails within H3K4me3/3xR-A-NCP.
DOI: https://doi.org/10.7554/eLife.31481.026
**Figure supplement 3.** Secondary structure analysis of modified/mutated H3 tails within the NCP.
DOI: https://doi.org/10.7554/eLife.31481.027
**Figure supplement 4.** Simulation time course of H3 tails within 4xK-Q-NCP.
DOI: https://doi.org/10.7554/eLife.31481.028
**Figure supplement 5.** Simulation time course of H3 tails within H3K4me3/4xK-Q-NCP.
DOI: https://doi.org/10.7554/eLife.31481.029
**Figure supplement 6.** H3 tail distribution in 4XK-Q- and H3K4me3/4xK-Q-NCP.
DOI: https://doi.org/10.7554/eLife.31481.030
**Figure supplement 7.** Difference in solvent-accessible surface area between acetylated lysine and glutamine.
DOI: https://doi.org/10.7554/eLife.31481.031

Sequential $^1$H-$^{15}$N HSQC spectra were recorded on $^{15}$N-PHD upon titration of H3K$_C$4me3/4xK-Q-, H3K$_C$4me3/3xR-A-, or H3K$_C$4me3/phos-NCP (*Figure 8A*, and *Figure 8—figure supplement 1–3*). Compared to titration of H3K$_C$4me3-NCP containing no additional modifications, the same set of residues is perturbed, and the resonances track along the same trajectories. However, at equivalent concentrations of NCP, PHD finger resonances have progressed farther towards the H3K$_C$4me3-Tail-bound state in binding to H3K$_C$4me3/4xK-Q-, H3K$_C$4me3/3xR-A-, and H3K$_C$4me3/phos-NCP as compared to binding to H3K$_C$4me3-NCP. This indicates that the PHD finger is binding the H3 tail with the same molecular mechanism between all NCP samples, but that when additional tail residues are modified or mutated, the observed binding affinity is higher. Failure to reach saturation precludes the ability to determine exact $K_d$ values; however, plotting the CSPs as a function of concentration of the H3K$_C$4me3 mark indicates that the binding trends in the order of the weakest to tightest binding to H3K$_C$4me3-NCP, followed by H3K$_C$4me3/4xK-Q- and H3K$_C$4me3/phos-NCP, with the tightest binding to H3K$_C$4me3/3xR-A-NCP (*Figure 8B*, *Figure 8—figure supplement 4*). Thus, neutralizing the charge of basic residues or introducing negative charges results in an increase in accessibility to PHD finger binding. This is consistent with the smaller computed binding energy observed between the H3 tail and nucleosomal DNA in the MD simulations of H3K4me3/quadAc-, H3K4me3/4xK-Q- and H3K4me3/3xR-A-NCP (see above), and the computed binding energies trend in the same order as the experimental data (see *Table 1*). Neutralization of arginine appears to have a greater effect than neutralization of lysine, especially when taking into account that only 3 arginine but four lysine residues were mutated. This is consistent with the computed greater favorable energetic contribution of the arginine than lysine residues (*Figure 7C*).

It is clear that altering the charge of several H3 tail residues is not sufficient to fully release the H3 tail from the nucleosomal DNA and recover the full binding potential of the PHD finger-H3K$_C$4me3 interaction. This further supports that many contacts along the length of the tail, albeit dominated by the basic residues, are

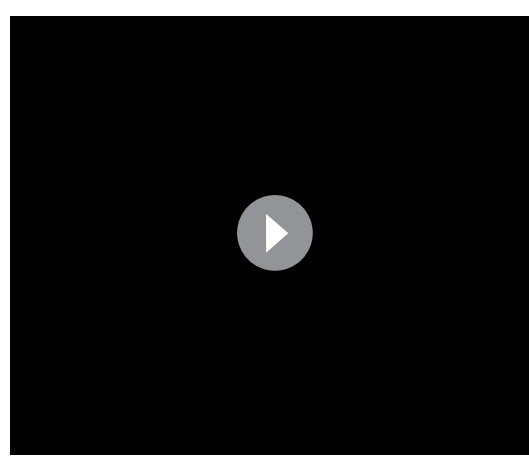

**Video 3.** End states of histone H3 tails from quadAc-NCP simulations. Protein backbone arrangement is shown in dark blue, while sidechains are represented as orange sticks. Sidechains interact with both DNA backbones (cyan and turquoise), as well as within both the major and minor grooves (grey), through a variety of conformations.
DOI: https://doi.org/10.7554/eLife.31481.032

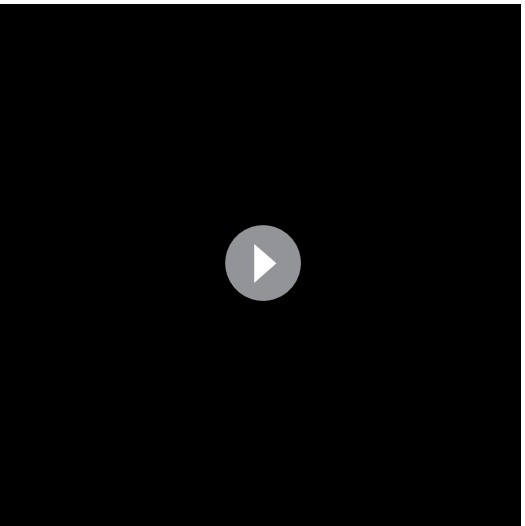

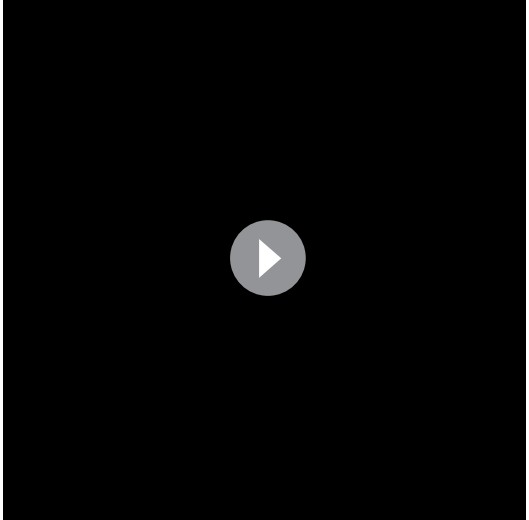

**Video 4.** End states of the histone H3 tails from the H3K4me3/3xR-A-NCP simulations. Protein arrangement is shown in dark blue, while sidechains are represented as orange sticks. Sidechains interact with both DNA backbones (cyan and turquoise), as well as within both the major and minor grooves (grey), through a variety of conformations.
DOI: https://doi.org/10.7554/eLife.31481.033

**Video 5.** End states of the histone H3 tails from 4xK-Q-NCP simulations. Protein backbone arrangement is shown in dark blue, while sidechains are represented as orange sticks. Sidechains interact with both DNA backbones (cyan and turquoise), as well as within both the major and minor grooves (grey), through a variety of conformations.
DOI: https://doi.org/10.7554/eLife.31481.034

involved in a high avidity interaction with the nucleosomal DNA.

## Discussion

In this study, we find that the nucleosome inhibits binding of the BPTF PHD finger to the methylated histone H3 tail. This adds to a growing body of evidence that the nucleosome context can have a significant effect on chromatin signaling events, including histone tail binding and modification (*Stützer et al., 2016*; *Wang and Hayes, 2007*; *Munari et al., 2012*; *Gatchalian et al., 2017*). We show that inhibition of PHD finger binding is due to the conformation of the tail in the context of the nucleosome core particle. Our data indicate a robust, but conformationally heterogeneous interaction of the H3 tail with the nucleosome core, driven by contacts with DNA (*Figure 9A*). The interaction with DNA is competitive with respect to PHD finger binding. As this collapsed conformation is favored under physiological conditions, this inhibits association of the PHD finger with the H3 tail in the context of the nucleosome (*Figure 9B*).

The classical model of the nucleosome depicts the tails as extended and accessible to binding. Indeed, the tails are highly susceptible to protease degradation and generally not resolved in crystal structures, and NMR data

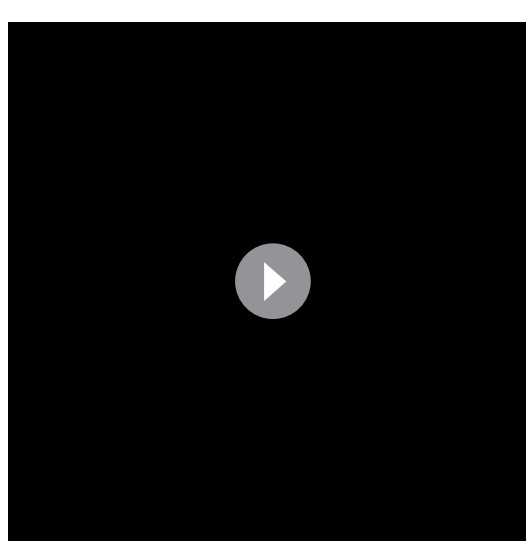

**Video 6.** End states of the histone H3 tails from the H3K4me3/4xK-Q-NCP simulations. Protein arrangement is shown in dark blue, while sidechains are represented as orange sticks. Sidechains interact with both DNA backbones (cyan and turquoise), as well as within the major and minor grooves (grey), through a variety of conformations.
DOI: https://doi.org/10.7554/eLife.31481.035

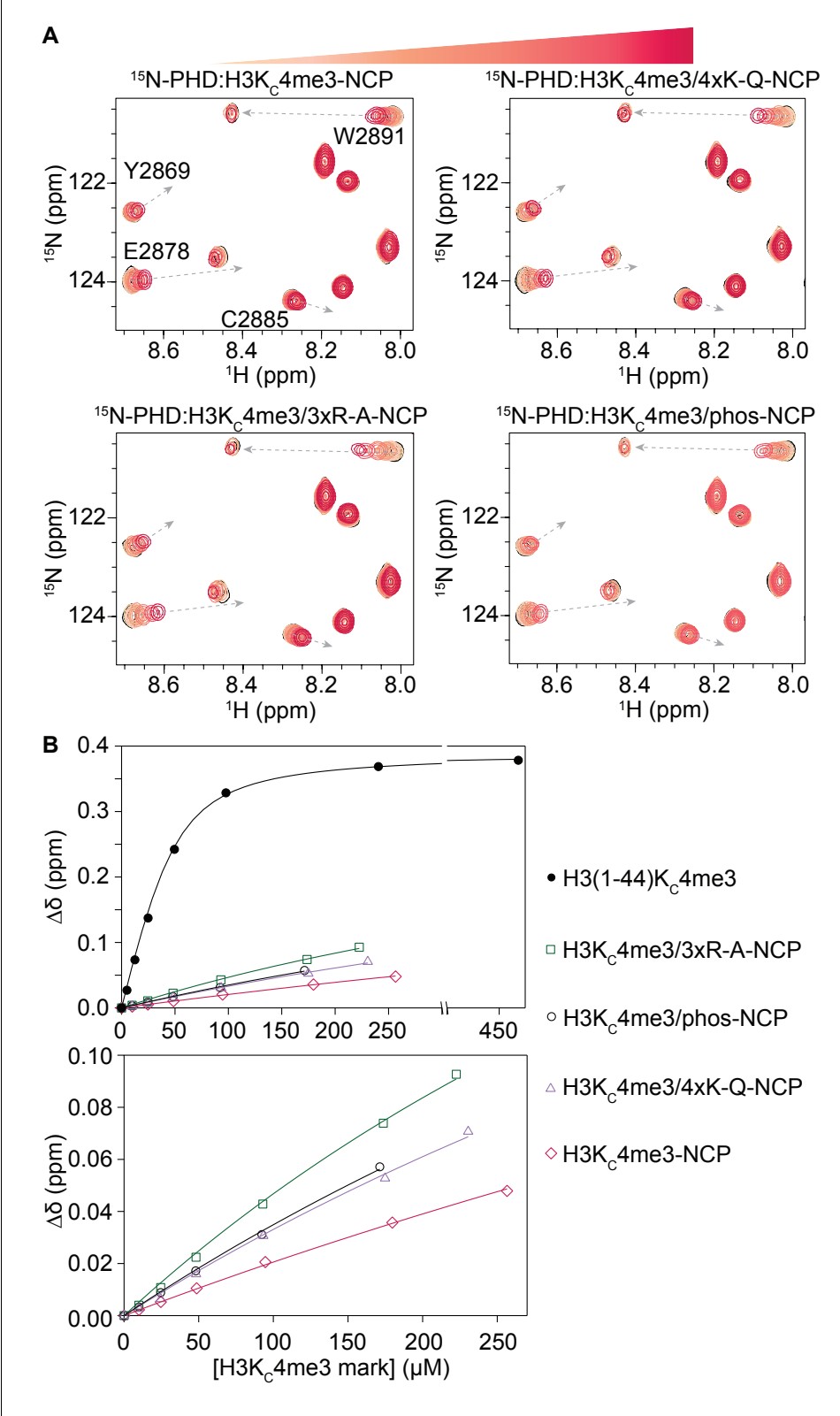

**Figure 8.** Modification of H3 tail residues remote from the binding interface with the BPTF PHD finger affects PHD finger binding. (A) Overlay of a region of the ${}^1$H-${}^{15}$N TROSY HSQC spectra of 50 µM ${}^{15}$N-BPTF PHD upon titration of H3K$_C$4me3-NCP containing no additional modifications, four lysine mutated to glutamine as acetyl-lysine mimetics, three arginine mutated to alanine, or phosphorylated serine on the H3 tail. Spectra are color-coded according to ligand concentration, with apo spectra in black and shades of salmon for increasing concentrations of ligand. Spectra were

*Figure 8 continued on next page*

*Figure 8 continued*

collected at molar ratios (PHD:H3K$_C$4me3 mark) of 1:0, 1:0.2, 1:0.5, 1:1, 1:2 and 1:4 and additionally 1:6, 1:5.5, and 1:5.4 for H3K$_C$4me3-, H3K$_C$4me3/4xK-Q-, and H3K$_C$4me3/3xR-A-NCP. Dashed, grey arrows track the trajectory between the apo and bound states for the H3K$_C$4me3-Tail titration and are shown as visual aids for titrations with NCP, which are predicted to have the same end state. (B) CSPs (Δδ) are plotted for each H3K$_C$4me3-NCP titration in (A) for the representative residue R2887 as a function of the concentration of the H3K$_C$4me3 mark. The H3K$_C$4me3-Tail titration is plotted for comparison.

DOI: https://doi.org/10.7554/eLife.31481.036

The following figure supplements are available for figure 8:

**Figure supplement 1.** Full spectra of NCP titrations into [15]N-BPTF PHD.
DOI: https://doi.org/10.7554/eLife.31481.037
**Figure supplement 2.** ESI data for histone constructs.
DOI: https://doi.org/10.7554/eLife.31481.038
**Figure supplement 3.** Gel characterization of NCP samples.
DOI: https://doi.org/10.7554/eLife.31481.039
**Figure supplement 4.** Modification of H3 tail residues outside of the BPTF PHD finger binding region affects the H3/PHD interaction.
DOI: https://doi.org/10.7554/eLife.31481.040

indicates substantial conformational dynamics within the tails (*Zhou et al., 2012*; *Böhm and Crane-Robinson, 1984*; *Rosenberg et al., 1986*; *Luger et al., 1997*; *Gao et al., 2013*). However, many biochemical and biophysical studies have also suggested that the histone tails interact with DNA and have an effect on nucleosome stability. Molecular dynamics studies also widely show that the histone tails collapse onto DNA during the course of a simulation (*Li and Kono, 2016*; *Shaytan et al., 2016*; *Ikebe et al., 2016*; *Erler et al., 2014*; *Bowerman and Wereszczynski, 2016*). Moreover, a CHIP-exo study suggests that the H3 tails associate with linker DNA *in vivo* (*Rhee et al., 2014*). Here we develop a model that is consistent with both (*Figure 9B*). We find that the H3 tails associate robustly with DNA in the context of the nucleosome, but with substantial conformational flexibility. Notably, DNA binding is independent of the presence of available linker DNA, as we find here that it also robustly associates with the core DNA. Our simulations reveal that there exists a plethora of energetically similar but structurally heterogeneous DNA-bound states of the H3 tail. Though simulations show that these binding modes remain highly stable up to the 100ns timescale, the fact that there is only a single peak observed for each residue by NMR suggests that movement between these collapsed conformations is fast on the NMR timescale. Fast exchange on the NMR timescale indicates dynamic transitions on the order of sub-microseconds. Movement between collapsed conformations could involve partially or fully released/extended conformations as intermediates or could take the form of a 'tethered diffusion' of the H3 tail along the surface of the core DNA.

Given current computing power, it is highly unlikely that individual MD simulations can rigorously sample a representative region of the H3 tail ensemble when bound to the NCP. These findings support previous studies, which showed that multiple trajectories must be collected and/or enhanced sampling approaches implemented in order to robustly simulate the bound state of the H3 tails (*Li and Kono, 2016*), and that caution should be exercised when drawing conclusions regarding the role of the tails in NCP stability and dynamics. The equilibrium of states is likely determined by a balance of the extended and collapsed modes, the former of which is driven by conformational entropy and the latter of which is driven by the enthalpy of tail-DNA interactions. At first glance, it may seem that the extended state should dominate due to its high flexibility and inherently large entropy. However, our results suggest that the collapsed tail state is actually more favorable. Specifically, our MM-GBSA calculations indicate that there is a significant enthalpic advantage to compacted configurations due to strong Coulomb interactions that overwhelm the loss of configurational entropy upon tail collapse. Furthermore, the large variability in compacted tail configurations observed in our simulation trajectories suggests that the collapsed state likely has a significant configurational entropy of its own, which may also alleviate some of the entropic cost of compaction.

Our results align with a recent study by *Stützer et al., 2016*. on the *X. laevis* 187bp-nucleosome. Similar to what we show here with inhibition of PHD finger binding, they found that acetylation and methylation of the H3 tail by a subset of histone acetyltransferases (Gcn5, p300, CBP) and methyltransferases (G9a, PRC2, PRMT5, Set7/9), respectively, is inhibited by the nucleosome. Using NMR, they determined this to be due to a transient interaction of the tail with linker DNA. We suggest a

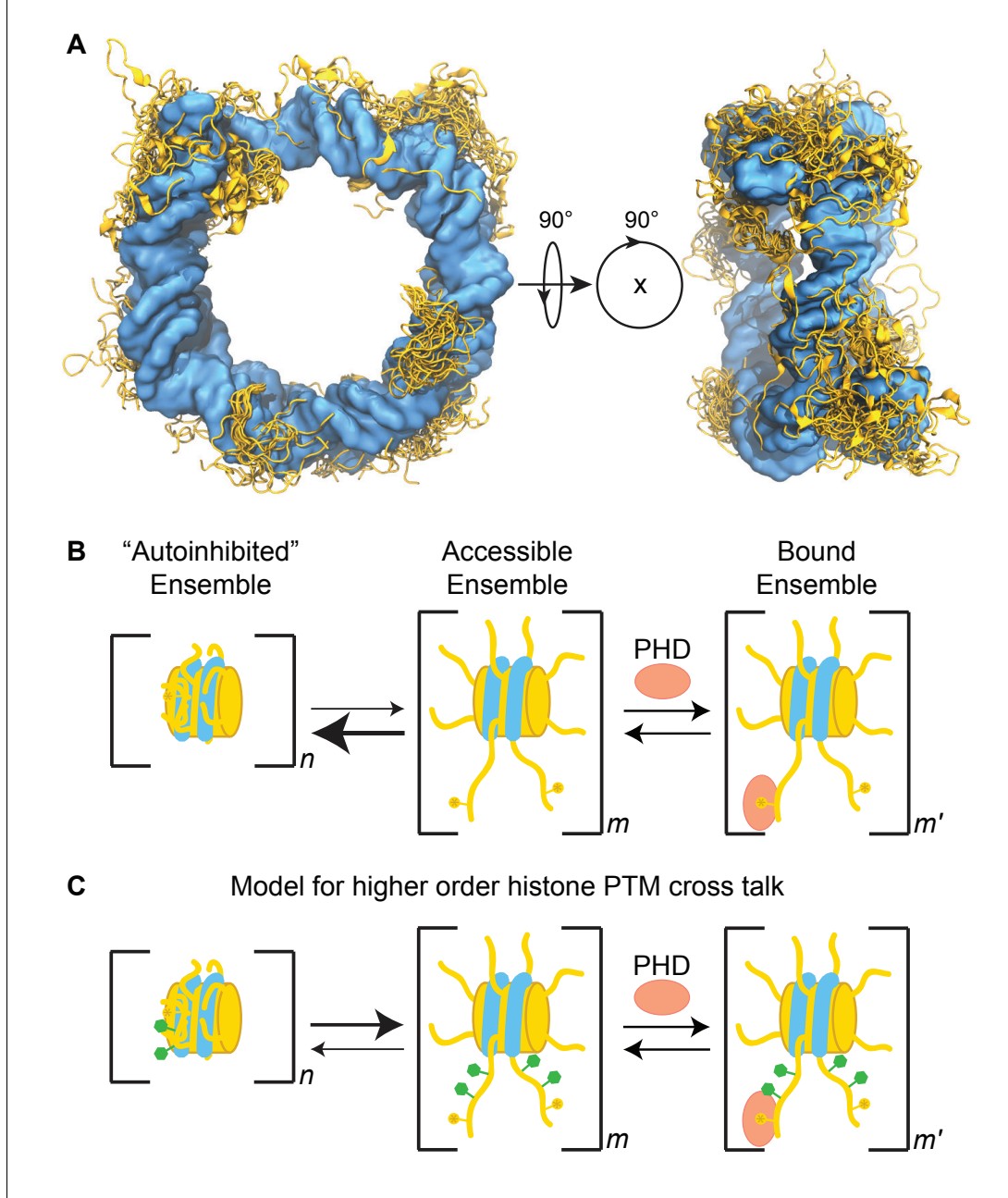

**Figure 9.** Model of the H3 tail conformational ensemble and effect on PHD finger binding. (**A**) Conglomeration of histone tail conformations (gold) from all simulations of the unmodified NCP compacted upon the core DNA surface (blue). (**B**) Cartoon model depicting the simplified linked equilibria between the collapsed, autoinhibited state (favored) and extended, accessible state that is permissive to PHD finger binding. The extended and collapsed states are ensembles of conformations as denoted by n, m, and m'. Note that each histone tail is likely in equilibrium between collapsed, partially extended, and extended conformations. We do not intend to indicate cooperativity between the equilibria of individual tails as this is unknown; distinct ensembles are depicted only for simplicity. We expect that the energetics of these ensembles may be perturbed by a number of nuclear factors, including histone PTMs. (**C**) The data provided predicts that higher order PTM crosstalk, or the modification of residues remote from the region directly recognized by the histone effector domain, can perturb the energetics of the system, thereby shifting the conformational pre-equilibrium and altering the observed binding affinity. Specifically, the neutralization of lysine or arginine side chains or the addition of negatively charged groups such as by phosphorylation of serine or threonine side chains result in a tighter observed binding affinity.
DOI: https://doi.org/10.7554/eLife.31481.041

robust yet dynamic interaction not only with linker DNA but also with DNA in the nucleosome core. It is likely that all histone tails are associated in some manner with nucleosomal DNA. An NMR analysis of the histone H4 tail found that the small basic patch within the tail (residues 16–20) is associated with the nucleosome core, as evidenced by the complete lack of signal in an $^1$H-$^{15}$N HSQC (*Zhou et al., 2012*). The contrast to observable signals for H3 could represent either a different exchange regime between states or even a single stable state for the H4 tail, as suggested by a recent computational study (*Erler et al., 2014*). In addition, a number of biochemical studies have shown through chemical reactivity and cross-linking that the N- and C-terminal tails are associated with the nucleosome core (for a review see [*Pepenella et al., 2014*]). This collapsed conformation presents a much more complex binding interface to histone effector domains than would an extended tail, and many effectors will likely be inhibited as seen here with the BPTF PHD finger. In fact, studies of fluorescein 5-maleimide reactivity towards single cysteine mutants of the H2B tail in the context of the nucleosome suggest that effector domain binding should be 10- to 50-fold weaker than binding to an H2B tail peptide (*Wang and Hayes, 2007*). A modest decrease in binding to the H3K9me3-NCP as compared to peptide was also observed for the chromodomain containing HP1β (*Munari et al., 2012*). This nucleosomal inhibition is less than that observed here for the BPTF PHD finger, but regions flanking the chromodomain are thought to interact with DNA, which could somewhat counter the inhibition. Another recent study reported on binding of the tandem CHD4 PHD fingers to the histone H3 tail unmodified at lysine 4[43]. Similar to our results, the individual CHD4 PHD fingers bound too weakly to the NCP to accurately measure binding affinity. Though the linked PHD fingers also bound more weakly to the nucleosome than to histone peptides, there was only a ~ 6 × difference in affinity (*Gatchalian et al., 2017*). The increase in affinity for the dual domain as compared to the individual PHD fingers is purportedly due to the multivalent activity of the tandem PHD fingers as well as contributions to histone binding by the region linking the two PHD fingers (*Gatchalian et al., 2017*).

There are many mechanisms by which the inhibitory conformations of the histone tails could be regulated. We show here that neutralization of lysine or arginine along the H3 tail, as occurs with several types of lysine acylation and arginine citrullination, weakens the association with DNA and promotes PHD finger binding. We also observe an increase of binding upon phosphorylation of serine. This is in agreement with recent studies showing that acetylation or phosphorylation of H3 tails promotes subsequent additional modification of the H3 tail (*Stützer et al., 2016*). Notably, there are many additional modifications including ADP-ribosylation and certain acylations that not only neutralize lysine, but add negative charge to the side-chain. In addition, modification by the small proteins ubiquitin and SUMO would likely lead to large steric hindrance. This suggests that cross-talk between histone PTMs is not only mediated through effector domains themselves (i.e. recognition of multiple PTMs by multiple domains, or antagonism between neighboring PTMs by direct inhibition of binding) but can also be mediated by the integrated effects of PTM recognition and nucleosome structure (*Figure 9C*). It is also possible that DNA-binding domains could displace the histone tail and increase accessibility. There are now several examples of histone effector domains that also harbor DNA-binding ability and have been found to bind tighter to nucleosomes than to histone peptides alone (*Charier et al., 2004*; *Kim et al., 2010*; *Qiu et al., 2012*; *Eidahl et al., 2013*; *Musselman et al., 2013*; *van Nuland et al., 2013*; *Savitsky et al., 2016*; *Miller et al., 2016*; *Morrison et al., 2017*). This could result from a combination of additional contacts with the nucleosome and displacement of the histone tail. A similar effect could be seen with adjacent DNA binding domains as suggested previously (*Pilotto et al., 2015*).

Altogether our data not only supports a model of the nucleosome where the tails are collapsed rather than extended, but also suggests many higher order regulatory mechanisms within chromatin signaling cascades, and highlights the importance of understanding the molecular mechanism of effector complexes at the nucleosome.

## Materials and methods

### BPTF PHD finger construct

A codon optimized BPTF PHD finger gene fragment (residues 2865–2924 of UniProt entry Q12830) was obtained from Integrated DNA Technologies (IDT, Coralville, IA) and cloned into the pDEST15

vector using Invitrogen Gateway recombination cloning technology (ThermoFisher Scientific, Waltham, MA) with an engineered N-terminal PreScission Protease cleavage site.

BL21 (DE3) Chemically Competent *E. coli* (ThermoFisher Scientific or New England Biolabs, Ipswich, MA) were used for expression. *E. coli* was grown in LB media or M9 minimal media supplemented with vitamin (Centrum, New York City, NY), 1 g L$^{-1}$ $^{15}$NH$_4$Cl, and 5 g L$^{-1}$ D-glucose to produce unlabelled or $^{15}$N-isotopically enriched protein, respectively. Media was supplemented with 100–200 µM ZnCl$_2$. Bacteria was grown to an OD$_{600}$ ~1.0 and induced with 0.3 mM IPTG at 18°C for 16–20 hr.

For purification of the GST-fusion PHD finger, cells were lysed in 20 mM Tris pH 7.5, 500 mM NaCl, 3 mM DTT, 0.5% Triton X-100, 0.5 mg mL$^{-1}$ lysozyme with DNaseI and Pierce EDTA-free Protease Inhibitor Tablets (ThermoFisher Scientific) using an Avestin (Canada) EmulsiFlex or sonication. The soluble portion of the lysate was incubated with glutathione agarose resin (ThermoFisher Scientific) and washed extensively with buffer (20 mM MOPS pH 7.0, 150 mM KCl, 1 mM DTT). Samples were cleaved from the GST tag overnight with PreScission Protease and further purified using anion exchange (Source 15Q, GE Healthcare Life Sciences, Pittsburgh, PA) and size exclusion chromatography (Superdex 75 10/300, GE Healthcare Life Sciences). The final buffer for all samples was 20 mM MOPS pH 7.0, 150 mM KCl, and 1 mM DTT. Note that 20 mM MOPS buffers were adjusted to pH 7 by adding 7 mM NaOH. The concentrations of PHD finger samples were determined via UV-vis spectroscopy using the absorbance (calculated $\varepsilon_{280}$ = 12,950 M$^{-1}$cm$^{-1}$).

## Histone purification and NCP reconstitution

Unmodified human histones (H2A.1 uniprot accession P0C0S8, H2B.1C uniprot accession P62807, H4 uniprot accession P62805 with T71C) were expressed out of pET3a vectors. A codon-optimized version of H3.2 (uniprot accession Q71DI3) was obtained from IDT and cloned into pET3a. Codon and non codon-optimized versions of H3 were made with a cysteine-free background (C110A), and the non codon-optimized version also carries the common G102A mutation. The Q5 mutagenesis kit (New England Biolabs) was used to introduce additional mutations. The H3 mutants generated were: 1) K4C, 2) K4C/K14Q/K18Q/K23Q/K27Q (referred to as H3K4C/4xK-Q), and 3) K4C/R8A/R17A/R26A (referred to as H3K4C/3xR-A). The K4C mutation was made to the non codon-optimized H3 and the K4C/K14Q/K18Q/K23Q/K27Q and K4C/R8A/R17A/R26A mutations were made to the codon-optimized H3. Rosetta 2 (DE3) pLysS (Novagen, Burlington, MA) or BL21 (DE3) (New England Biolabs) chemically competent *E. coli* were used for expression. Unlabelled growths were carried out in either LB or M9 media and were induced at OD$_{600}$ ~ 0.4 with 0.2 mM (for H4) or 0.4 mM (for H2A, H2B, and H3) IPTG for 3–4 hr. $^{15}$N- and $^{13}$C/$^{15}$N-isotopically enriched H3 was grown in M9 minimal media supplemented with vitamin (Centrum), 1 g L$^{-1}$ $^{15}$NH$_4$Cl, and 5 g L$^{-1}$ unlabelled or 3 g L$^{-1}$ $^{13}$C D-glucose. Histones were extracted from inclusion bodies following (*Qiu et al., 2012*) and purified via ion exchange chromatography.

The trimethyl lysine analogue was installed using alkylation of H3K4C following (*Simon, 2010*). Briefly, H3K4C was dissolved at 10 mg/mL in alkylation buffer (4M guanidine hydrochloride, 1M HEPES pH 7.8, 10 mM DL-methionine) with 20 mM fresh DTT by incubating at 37°C for 1 hr. (2-Bromoethyl) trimethyl ammonium bromide was added as the alkylating agent at 100 mg/mL and incubated at 50°C for 2.5 hr. An additional 10 mM of fresh DTT was added, followed by another 2.5 hr incubation. Each 1 mL reaction was quenched with 50 µL β-mercaptoethanol, and H3K$_C$4me3 was de-salted using a PD10 column (GE Healthcare Life Sciences) and dialyzed against 2 mM β-mercaptoethanol in H$_2$O.

Unmodified, H3K$_C$4me3, H3K$_C$4me3/4xK-Q, and H3K$_C$4me3/3xR-A octamers were prepared as described in (*Qiu et al., 2012*). Briefly, equimolar ratios of histones were mixed in 20 mM Tris pH 7.5, 6M Guanidine HCl, 10 mM DTT, dialyzed into 20 mM Tris pH 7.5, 2M KCl, 1 mM EDTA, 5 mM β-ME, and purified over a sephacryl S-200 column (GE Healthcare Life Sciences) via FPLC.

A plasmid containing 32 repeats of the 147 bp Widom 601 sequence (ATCGAGAATCCCGG TGCCGAGGCCGCTCAATTGGTCGTAGACAGCTCTAGCACCGCTTAAACGCACGTACGCGCTG TCCCCCGCGTTTTAACCGCCAAGGGGATTACTCCCTAGTCTCCAGGCACGTGTCAGATATATACA TCCGAT) was amplified in *E. coli* and purified via alkyline lysis methods, largely as outlined in (*Qiu et al., 2012*). The 601 repeats were released by cleavage with EcoRV and purified from parent plasmid by polyethylene glycol precipitation, and further purified over a source 15Q column (GE Healthcare Life Sciences) via FPLC.

Unmodified, H3K$_C$4me3, H3K$_C$4me3/4xK-Q, and H3K$_C$4me3/3xR-A NCPs were reconstituted with the 147 bp Widom 601 sequence via desalting methods (*Qiu et al., 2012*). Briefly, octamer and 601 DNA were mixed at a 1:1 molar ratio and desalted using a linear gradient from 2M to 150 mM KCl over ~48 hr. NCPs were heat-shocked at 37°C for 30 min to obtain uniform positioning and then purified using a 10–40% sucrose gradient. Proper nucleosome formation was confirmed via native polyacrylamide gel electrophoresis and by the sucrose gradient profile.

Nucleosome concentrations were determined via UV-vis spectroscopy using the absorbance from the 601 DNA (calculated $\varepsilon_{260}$ = 2,312,300.9 M$^{-1}$cm$^{-1}$). Before measuring the concentration, samples were diluted into 2M KCl in order to promote nucleosome disassembly for more accurate concentration determination.

In order to produce tailless NCPs (tlNCPs), reconstituted NCPs were treated with TPCK Trypsin immobilized on magnetic beads (Takara Bio, Japan). NCPs were incubated with the beads at room temperature for 30 min in 0.5xTE (which resulted in little digestion) and then for 70 min with 75 mM KCl (see *Figure 3—figure supplement 2* for time points). The digestion and final sample resembled (*Ausio et al., 1989*).

The H3K$_C$4me3-NCP sample was treated with Aurora B kinase in order to phosphorylate serine 10 and 28 on H3 (H3K$_C$4me3/S10ph/S28ph-NCP, referred to as H3K$_C$4me3/phos-NCP). After titrating the H3K$_C$4me3-NCP sample into $^{15}$N-BPTF PHD, the H3K$_C$4me3-NCP sample was recovered and ~1 mM EDTA was added to unfold the PHD finger. This recovered NCP was treated with recombinant human Aurora B (AurB) protein (abcam product ab51435, Cambridge, MA). The NCP was treated with 3 μg of AurB for 4.2 hr at 30°C (4.4 mM MgCl$_2$, 0.6 mM EGTA, 0.7 mM EDTA, 1.5 mM β-glycerophosphate, 2 mM ATP, 0.6 mM benzamidine, 20 mM MOPS pH 7, 150 mM KCl, 0.7 mM DTT). Following the incubation, an additional 4 mM EDTA was added to chelate the free Mg$^{2+}$. The treated NCP was separated from PHD finger and kinase reaction buffer components by purifying it over a Superdex 75 10/300 column (GE Healthcare Life Sciences). Phosphorylation of H3K$_C$4me3 was confirmed via an 18% SDS-PAGE gel prepared using Zn$^{2+}$-Phos-tag acrylamide AAL-107 (Wako Pure Chemical Industries, Japan, see *Figure 8—figure supplement 3*). The gel was prepared and run according to recommendation as outlined in the Wako Pure Chemical Industries manual for Zn$^{2+}$-Phos-tag for SDS-PAGE using a neutral-pH buffer system, using 50 μM Zn$^{2+}$-Phos-tag acrylamide AAL-107, including ZnCl$_2$ at a final concentration of 0.3 mM in all gel samples (including the ladder and empty wells), and running the gel on ice.

## Histone peptides

The histone peptide H3(1–10)K4me3 was obtained from Anaspec (Fremont, CA). Concentrated peptide stocks were prepared in H$_2$O based on the weight provided by the company. The pH of stock solutions was adjusted to pH ~7 with NaOH.

The H3(1–44) construct (ARTKQTARKS TGGKAPRKQL ATKAARKSAP ATGGVKKPHR YRPG) was expressed out of pET3a after inserting two stop codons (TAATAA) between codons for residues 44 and 45 into the pET3a plasmid containing codon-optimized H3 using the Q5 mutagenesis kit (New England Biolabs). This kit was also used to generate the K4C mutant. Unlabeled, $^{15}$N- or $^{13}$C/$^{15}$N-isotopically enriched H3(1–44) or H3K4C(1-44) were expressed in BL21 (DE3) chemically competent *E. coli* (New England Biolabs) in the same manner as for the full length H3 (see above), except that cells were grown to OD$_{600}$ ~1.0 prior to induction. To purify H3(1–44) constructs, cells were lysed in 50 mM Tris pH 7.5, 100 mM NaCl, 1 mM benzamidine, 2 mM EDTA, 0.5% Triton X-100, 0.5 mg mL$^{-1}$ lysozyme with DnaseI and Pierce Protease Inhibitor Tablets (ThermoFisher Scientific) using an Avestin EmulsiFlex. The soluble portion of the lysate was purified using cation exchange resin and size exclusion chromatography (Superdex 30, GE Healthcare Life Sciences) in 50 mM KPi pH 7, 50 mM KCl, 0.5 mM EDTA. Samples were then desalted by purifying over a Superdex 75 column (GE Healthcare Life Sciences) and lyophilized. Final samples were prepared in 20 mM MOPS pH 7.0, 150 mM KCl, and 1 mM DTT (and with 1 mM EDTA for samples that would not be used with the BPTF PHD finger). Note that 20 mM MOPS buffers were adjusted to pH 7 by adding 7 mM NaOH. The H3(1–44) K4C sample was alkylated following the same protocol elaborated above for the full length H3K4C, and the proper alkylation (trimethyl lysine analogue) was confirmed via ESI mass spectrometry (see *Figure 1—figure supplement 3*). The concentrations of H3(1–44)K$_C$4me3 stocks were determined via UV-vis spectroscopy using the absorbance from the native tyrosine Y41 (using the calculated $\varepsilon_{280}$ = 1490 M$^{-1}$cm$^{-1}$).

## Mass spectrometry on histone samples

ESI mass spectrometry was used to analyze the full length H3 histones and histone tails, to confirm the removal of the N-terminal methionine, check that there was no carbamylation, and confirm proper alkylation of the MLAs. A Waters Q-Tof Premier instrument (Milford, MA) was used with positive electrospray ionization (ESI). Samples were diluted 1:2 or 1:4 in water/acetonitrile (1:1) with 0.1% formic acid. The acquisition and deconvolution software used during data collection and analysis were MassLynx and MaxEnt, respectively.

## DNA samples

Oligos for use in NMR studies were obtained from IDT with the following sequences: 5'-CTCAA TTGGTCGTAGACAGCT-3' and 5'-AGCTGTCTACGACCAATTGAG-3'. Double stranded oligos were annealed at a concentration of 350 µM by heating to 94°C for 10 min followed by a slow cooling to room temperature. The annealed duplexes were purified by size exclusion chromatography (Superdex 75 10/300, GE Healthcare Life Sciences) in NMR buffer (20 mM MOPS pH 7.0, 150 mM KCl, 1 mM DTT, and 1 mM EDTA in samples not used with the PHD finger) and concentrated. Concentration of DNA stocks was determined via UV-vis spectroscopy using the extinction coefficient predicted by the IDT Biophysics UV spectrum tool for duplex DNA at 260 nm (http://biophysics.idtdna. com/cgi-bin/uvCalculator.cgi), which is 333,804.5 $M^{-1}cm^{-1}$ for this DNA.

## NMR spectroscopy and data analysis with the BPTF PHD finger

Assignments for the BPTF PHD finger were transferred from (*Li et al., 2006*). Data was compared between buffer conditions and temperature titrations were performed to ensure proper transfer of assignments.

Titrations of H3 tail peptides and NCPs into $^{15}$N-PHD were carried out by collecting $^1$H-$^{15}$N HSQC spectra on $^{15}$N-PHD in the apo state and with increasing concentrations of substrate, using TROSY with NCP substrates. $^{15}$N-PHD samples were at 50 µM in 20 mM MOPS pH 7, 150 mM KCl, 1 mM DTT, and 7% $D_2O$. Titrations were collected at PHD:peptide ratios of 1:0, 1:0.1, 1:0.25, 1:0.5, 1:1, 1:2, 1:5, and additionally 1:10 with H3(1–44)$K_C$4me3. For titrations with H3 tail peptides prebound to DNA the peptide and DNA were mixed at a ratio of 1:2, and this stock was titrated into the $^{15}$N-PHD at the same PHD:peptide ratios (without the 1:0.1 point for H3(1–44)$K_C$4me3). Titrations with H3$K_C$4me3-, H3$K_C$4me3/4xK-Q-, H3$K_C$4me3/3xR-A-, and H3$K_C$4me3/phos-NCP were collected at PHD:NCP ratios of 1:0, 1:0.1, 1:0.25, 1:0.5, 1:1, and 1:2 and additionally a final point at 1:3, 1:2.8, and 1:2.7 for H3$K_C$4me3-, H3$K_C$4me3/4xK-Q-, and H3$K_C$4me3/3xR-A-NCP. The NCPs were stable over the course of all titrations, as supported by native gels (see *Figure 8—figure supplement 3* for examples of gels). Data was collected at 37°C on an 800MHz Bruker (Billerica, MA) spectrometer with a cryogenic probe, collecting 24 scans for titrations with the H3 tail peptides and 32 scans for titrations with the NCPs.

Titration data were processed in NMRPipe (*Delaglio et al., 1995*) and analyzed using CcpNmr Analysis (*Vranken et al., 2005*). Binding curves were fit using a nonlinear least-squares analysis in Igor (Wavemetrics, Portland, OR) to a single-site binding model under ligand-depleted conditions:

$$\Delta\delta = \Delta\delta_{max}\left(([L]+[P]+K_d) - \sqrt{([L]+[P]+K_d)^2 - 4[P][L]}\right)/(2[P])$$

where [P] is the concentration of protein, [L] is the concentration of histone peptide or DNA, $\Delta\delta_{max}$ is the chemical shift difference at saturation. The combined chemical shift difference ($\Delta\delta$) at each point in the titration is calculated by:

$$\Delta\delta = \sqrt{(\Delta\delta_H)^2 + (0.154\Delta\delta_N)^2}$$

where $\Delta\delta_H$ and $\Delta\delta_N$ are the changes in the $^1$H and $^{15}$N chemical shift, respectively, at each titration point with respect to the apo chemical shifts. Binding curves were fit using two independent variables (ligand and protein concentrations, to account for dilution) for residues that were significantly perturbed upon binding. Residues were determined to be significantly perturbed if the $\Delta\delta$ was larger than the average plus one-half standard deviation of the $\Delta\delta$ values for all residues. (The relatively low cutoff of one-half standard deviation was chosen for the PHD finger because the binding interface of

such a small domain represents a relatively large portion of the domain.) Reported $K_d$ values were determined by fitting $K_d$ values for the significantly perturbed individual residues, calculating the average and standard deviation of the $K_d$ values for these residues, and removing residues with fit $K_d$ values not within the average ±two standard deviations (which was never more than a single residue). This resulted in using 14 residues for H3(1–44)$K_C$4me3 and 15 residues for H3(1–44)$K_C$4me3 pre-bound to DNA.

## NMR spectroscopy and data analysis with the H3 tail

Assignments on nucleosomes reconstituted using 167 or 187 bp 601 DNA have been published (*Zhou et al., 2012*; *Stützer et al., 2016*). We confirmed assignments with NCP reconstituted using 147 bp 601 DNA because assignments have not been published with this construct. To obtain backbone assignments for native H3 within the context of the NCP, HNCACB, CBCAcoNH, and HNCO spectra were collected on a 320 µM $^{13}C/^{15}N$-H3-NCP sample (i.e. 640 µM of H3 component) using a 600MHz Varian spectrometer at 45°C. Data was processed in NMRPipe (*Delaglio et al., 1995*) and analyzed using CcpNMR Analysis (*Vranken et al., 2005*). The HNCACB and CBCAcoNH were collected with 32 and 24 scans, respectively, and 56 and 36 complex increments in the $^{13}C$- and $^{15}N$-dimensions, respectively. The HNCO was collected with 8 scans and 36 complex increments in both the $^{13}C$- and $^{15}N$-dimensions, respectively. Temperature titration was used to transfer assignments to 25°C and 37°C. $^1H$-$^{15}N$ HSQC spectra collected on an H3$K_C$4me3-NCP sample were used to help confirm assignments of degenerate sections.

Backbone assignments were made for DNA-bound $^{13}C/^{15}N$-H3(1–44). The DNA-bound state was chosen to resolve degeneracy observed in the apo state. Data was collected on a sample containing 1.1 mM $^{13}C/^{15}N$-H3(1–44) and 1.4 mM DNA in 20 mM MOPS pH 7, 150 mM KCl, 1 mM EDTA, 8% $D_2O$ using a 500 Bruker spectrometer at 10°C. Assignments were made based on HNCACB (40 scans with 43 and 40 complex increments in the $^{13}C$- and $^{15}N$-dimensions, respectively) and HNco-CACB (8 scans with 40 complex increments in the $^{13}C$- and $^{15}N$-dimensions) spectra. Temperature and DNA titrations were used to transfer assignments between conditions.

$^1H$-$^{15}N$ HSQC spectra were collected on $^{15}N$-H3(1–44) and $^{15}N$-H3-NCP samples at a range of KCl and MgCl$_2$ concentrations and temperatures on an 800MHz Bruker spectrometer with cryogenic probe.

Titrations of 21 bp DNA (see above), NCP, and tlNCP into $^{15}N$-H3(1–44) were carried out by collecting sensitivity enhanced (SE) SOFAST $^1H$-$^{15}N$ HMQC spectra on $^{15}N$-H3(1–44) in the apo state and with increasing concentrations of substrate. Buffer conditions were 20 mM MOPS pH 7, 150 mM KCl, 1 mM DTT, and 1 mM EDTA. The DNA titration was repeated with 1 mM MgCl$_2$ and no EDTA to test for the effect of Mg$^{2+}$. Titrations were collected at H3(1–44):DNA ratios of 1:0, 1:0.1, 1:0.25, 1:0.5, 1:1, 1:2, and 1:4 and H3(1–44):NCP/tlNCP of roughly 1:0, 1:0.2, 1:0.5, 1:1, 1:2, and 1:3. The titrations were collected at 25°C on an 800MHz Bruker spectrometer with cryo probe.

Titration data were processed in NMRPipe (*Delaglio et al., 1995*) and analyzed using CcpNmr Analysis (*Vranken et al., 2005*). The composite chemical shift difference (Δδ) at each point in the titration is calculated by:

$$\Delta\delta = \sqrt{(\Delta\delta_H)^2 + (0.154\Delta\delta_N)^2}$$

where $\Delta\delta_H$ and $\Delta\delta_N$ are the changes in the $^1H$ and $^{15}N$ chemical shift, respectively, at each titration point with respect to the apo chemical shifts.

## Biolayer interferometry (BLI) data collection and analysis

An N-terminal biotin tag was added to the BPTF PHD finger for BLI experiments. The Q5 mutagenesis kit (New England Biolabs) was used to introduce an AviTag and an additional linker (GLNDIFEA QKIEWHEGSGS). The AviTag-PHD construct was grown and purified as described above, with the exception that this version of the BPTF PHD finger was not codon optimized and expression was done out of Rosetta 2(DE3)pLysS competent cells (Novagen). The AviTag-PHD construct was biotinylated in vivo by endogenous BirA (and confirmed via western blot, data not shown). Cells were grown in LB media, and 100 µM biotin was added at induction. All of the experiments were performed using biotin-PHD from a single protein prep so the population of biotinylated protein was constant between experiments.

Experiments were run using an Octet RED96 Biolayer Interferometry (BLI) System (Pall ForteBio, Menlo Park, CA). All samples were prepared in 20 mM MOPS pH 7, 150 mM KCl, 1 mM DTT, 0.2 mg/mL BSA (RPI albumin, bovine fraction V, molecular biology grade). Dip and Read Steptavidin Biosensors (Pall ForteBio) were used for all experiments and were hydrated in buffer for at least 30 min preceding data collection. Experiments were performed at 37°C in black 96-well plates (Greiner Bio-One, Monroe, NC), shaking at 1000 rpm. Data were collected at 10 Hz with a scheme of 10 min temperature pre-equilibration followed by 180 s buffer equilibration, 300 s biotin-PHD loading, 120 s buffer baseline, 300 s analyte association, and 300 s analyte dissociation steps. Two data sets were collected with H3(1–44)$K_C$4me3 as analyte. For one of the data sets, double referencing was performed. This consisted of the standard single reference sensor that was loaded with biotin-PHD with the association phase performed using buffer alone and additionally a second set of sensors, not loaded with biotin-PHD, that were run through the same protocol for all analyte concentrations to test for optical property changes and non-specific binding. As no difference was observed between either referencing, the second data set was obtained with only the single reference of biotin-PHD against buffer. H3(1–44)$K_C$4me3 analyte concentrations used were 20, 10, 5, 2.5, 1.3, and 0.6 μM, and the double referenced experiment additionally used 0.3 μM. Separate peptide dilutions were prepared for each experiment. Two data sets were also collected with H3$K_C$4me3-NCP as analyte. Note that double referencing was needed for the NCP as a large offset in response signal was seen independent of the PHD finger, likely due to changes in the optical properties of the solution. H3$K_C$4me3-NCP analyte concentrations used were 183, 21, and 10 μM for one experiment and 230, 23, and 11 μM for the other experiment.

Data were analyzed using the Octet Analysis software. Data were processed by subtracting the single or double reference data, aligning to the last 5–10 s of the baseline, and applying a Savitzky-Golay smoothing filter. Fast kinetics precluded the use of kinetic fits to determine binding affinity. Instead, equilibrium dissociation values were calculated by taking the average of the last 20 s of the association phase ($r_{eq}$) and plotting this against the analyte concentration ($x$). This curve was fit to obtain the $K_d$ value using a simple single-site binding equation:

$$r_{eq} = r_0 + (r_{max} - r_0)\frac{x}{K_d + x}$$

Data were fit using a nonlinear least-squares analysis in Igor (Wavemetrics).

## Molecular dynamics simulations

Simulations of nucleosomes containing six modification states of the H3 tail were performed: an unmodified tail (denoted un-NCP), H3K4me3 (H3K4me3-NCP), and H3K14,18,23,27ac (quadAc-NCP), H3K14,18,23,27ac (4xK-Q-NCP), a combination of H3K4me3 and H3K14,18,23,27Q (H3K4me3/4xK-Q-NCP), and a combination of H3K4me3 and H3R8,17,26A (H3K4me3/3xR-A-NCP). For each system, NCP simulations were initiated from three different initial H3 tail conformations: one based on the 1KX5 crystal structure, one in which the H3 tails were extended linearly from residues 1 to 40, and one *de novo* structure predicted with the MODELLER software package (*Shen and Sali, 2006*). In the MODELLER calculations, a set of 25 possible H3 tail structures was created and used to construct a collection of 625 potential NCP models by substituting one of the 25 generated tail states for each copy of H3 in the 1KX5 NCP. These 625 conformations were then energy minimized in an implicit solvent environment (*Onufriev et al., 2004*), and the initial conformation used for simulations was taken as the one with the most favorable energetics for the un-NCP. In all systems, the missing three residues of H2B ([1]PEP[3]) were extended linearly from the histone N-terminus using tLeap. By considering each of these three H3 tail conformations, simulations were conducted on a total of nine different systems.

Protein and DNA parameters were based on the Amber14SB and bsc1 forcefields (*Maier et al., 2015*; *Ivani et al., 2016*), and PTM parameters were previously determined by Papamokos et al. (*Papamokos et al., 2012*). All systems were neutralized and solvated in a TIP3P solution of 150 mM KCl (*Jorgensen et al., 1983*; *Joung and Cheatham, 2009*). Heavy atom masses were repartitioned to their associated H atoms so that, in conjunction with SHAKE restraints, a four fs time-step could be used (*Hopkins et al., 2015*; *Ryckaert et al., 1977*). Simulations were conducted in the CUDA-enable pmemd engine (v16) (*Salomon-Ferrer et al., 2013*; *Le Grand et al., 2013*), and five separate simulations for each combination of NCP system and tail conformation were performed, for a total

of ninety independent simulations. For each simulation, energy minimization was performed for 5000 steps with solute heavy atoms restrained by a 10 kcal/mol/Å$^2$ harmonic potential, followed by 5000 steps with no restraints. Then, each simulation was heated from 10 K to 300 K over 50 ps in the NVT ensemble with the restraints reinstated. Next, the restraints were gradually released over 250 ps in the NPT ensemble, using a Monte Carlo barostat with a target pressure of 1 atm and a relaxation time of 3 ps. Temperature was regulated using a Langevin thermostat (*Loncharich et al., 1992*) with a collision frequency of 3.0 ps$^{-1}$. Simulations were then conducted for 150 ns in the NPT ensemble with no restraints, yielding a cumulative sum of 13.5 μs of simulation time across all systems. Trajectories were recorded every 10 ps, and frames were visualized using VMD (*Humphrey et al., 1996*) and PyMol. Since 100 ns of simulation was required for the systems to equilibrate, only the last 50 ns of each simulation was used in the computational analysis (750 ns net for each NCP system).

## Simulation analysis

Interaction energies between the H3 tails and DNA in each system were determined from the sum of tail residue contributions to DNA binding according to an MM-GBSA (Molecular Mechanics Generalized Born Surface Area) analysis (*Miller et al., 2012*). Reported energies are the average of all results from each single simulation of a set (i.e., tail chemistry or initial tail conformation). Each tail within a simulation is considered a separate observation of the H3 tail ensemble, thereby producing two samples per simulation. Errors in the calculations are presented as the standard error of the mean from the simulation set, which corresponds to the standard deviations within the 10 samples of each initial tail conformation for a particular tail chemistry (a cumulative of 30 samples per tail chemistry).

Residue secondary structures were calculated using the DSSP algorithm (*Kabsch and Sander, 1983*), as implemented in cpptraj (*Roe and Cheatham, 2013*). For simplicity, secondary structures are reported as one of four categories: helix (DSSP types 'G', 'H', and 'I'), sheet ('E', 'B'), turn ('T'), or unstructured coil (none of the above). Total tail helicity or sheet values are reported as the average over all tail residues of each residue's percentage of frames in the respective structure. The global H3 tail structure was monitored through the radius of gyration ($R_g$) of the whole H3 histone such that reduced values in $R_g$ can be directly interpreted as tail compaction upon the core. Furthermore, the 3-D coordinates of heavy atoms in the tails are visualized as average occupancy within a grid of 1.0 Å$^3$ voxels, in a method similar to the counter-ion condensation observations made by Materese et al (*Materese et al., 2009*). Solvent exposure was calculated using the LPCO method, as implemented in cpptraj (*Weiser et al., 1999*). Lastly, a comparative analysis of the root mean square deviation (RMSD) of tail residues in the final frame of each simulation was conducted. In this analysis, translations and rotations were removed by least squares fitting the backbone of H3 core residues, and the subsequent RMSD values consider only the H3 tail residue backbone atoms.

## Acknowledgements

Work in the Musselman group is funded by an NSF CAREER Award (1452411). EAM is supported by an Arnold O. Beckman Postdoctoral Fellowship, and previously by an Iowa Cardiovascular Interdisciplinary Research Fellowship (T32HL007121). Work in the Wereszczynski group was supported by an NSF CAREER Award (1552743) and also by the National Institute of General Medical Sciences of the National Institutes of Health under award number R35GM119647. The content is solely the responsibility of the authors and does not necessarily represent the official views of the National Institutes of Health. This work used the Extreme Science and Engineering Discovery Environment (XSEDE [*Towns et al., 2014*]), which is supported by National Science Foundation Grant No. ACI-1053575. We would like to thank the Carver College of Medicine NMR facility, especially Dr. Liping Yu for implementing the SE-SOFAST HMQC. We would additionally like to thank the High Resolution Mass Spectrometry Facility (Office of the Vice-President for Research and Economic Development at the University of Iowa), especially Dr. Lynn Teesch. In addition, we would like to acknowledge use of BLI resources at the Carver College of Medicine's Protein Crystallography Facility at the University of Iowa, and acknowledge Nicholas Schnicker's training and advice with the BLI data collection. Additional thanks to Kathy Varzavand for preparing histones and 601 DNA, Drs. Karolin Luger and Michael Poirier for the gifts of the histone plasmids, and the UI CERT group for helpful discussions.

# Additional information

## Funding

| Funder | Grant reference number | Author |
|---|---|---|
| National Science Foundation | 1452411 | Catherine A Musselman |
| National Science Foundation | 1552743 | Samuel Bowerman Jeff Wereszczynski |
| Arnold and Mabel Beckman Foundation | Postdoctoral Fellowship | Emma A Morrison |
| National Institutes of Health | R35GM119647 | Jeff Wereszczynski |

The funders had no role in study design, data collection and interpretation, or the decision to submit the work for publication.

## Author contributions

Emma A Morrison, Conceptualization, Data curation, Formal analysis, Funding acquisition, Validation, Investigation, Visualization, Methodology, Writing—original draft, Writing—review and editing; Samuel Bowerman, Conceptualization, Data curation, Formal analysis, Validation, Investigation, Visualization, Methodology, Writing—original draft, Writing—review and editing; Kelli L Sylvers, Data curation, Formal analysis, Investigation, Visualization; Jeff Wereszczynski, Catherine A Musselman, Conceptualization, Data curation, Formal analysis, Supervision, Funding acquisition, Validation, Investigation, Visualization, Methodology, Writing—original draft, Project administration, Writing—review and editing

## Author ORCIDs

Emma A Morrison (iD) http://orcid.org/0000-0001-6722-7961
Samuel Bowerman (iD) http://orcid.org/0000-0003-0753-4294
Kelli L Sylvers (iD) http://orcid.org/0000-0003-0711-402X
Catherine A Musselman (iD) http://orcid.org/0000-0002-8356-7971

## Decision letter and Author response

Decision letter https://doi.org/10.7554/eLife.31481.044
Author response https://doi.org/10.7554/eLife.31481.045

# Additional files

## Supplementary files

• Transparent reporting form
DOI: https://doi.org/10.7554/eLife.31481.042

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
