## [Decision Letter]

[Editors’ note: this article was originally rejected after discussions between the reviewers, but the authors were invited to resubmit after an appeal against the decision.]

Thank you for submitting your work entitled "The conformation of the histone H3 tail inhibits association of the BPTF PHD finger with the nucleosome" for consideration by *eLife*. Your article has been reviewed by three peer reviewers, one of whom, Geeta J Narlikar is a member of our Board of Reviewing Editors and the evaluation has been overseen by a Senior Editor.

Our decision has been reached after consultation between the reviewers. Based on these discussions and the individual reviews below, we regret to inform you that your work will not be considered further for publication in *eLife*.

As you will see from the reviews appended below, the reviewers thought that the work has the potential to be impactful because quantifying the magnitude of any inhibitory effect of the nucleosome on reader binding will uncover the dynamic range of tail accessibility that can be tuned by other nucleosome modifiers. However, there were concerns that the work as such did not go sufficiently beyond the published work by Stutzer et al., 2016. Stutzer et al., have previously shown large inhibitory effects in the context of nucleosomes with linker DNA. The reviewers recognize that your work differs in that it assesses effects with core nucleosomes. The reviewers also agree that the data suggest that direct interactions between the H3 tails and core nucleosomal DNA restrict tail accessibility for reader domain binding. While this result is interesting, where the study has the potential for larger impact is from a direct comparison of the Kd of the PHD reader protein on nucleosomes vs. tail peptides. Here, the lack of saturation observed in the context of a nucleosome raised concerns about obtaining a reliable Kd. It is possible that alternative methods such as ITC could be used to compare the Kd on a nucleosome vs. peptides or higher concentrations of the PHD finger could be used in the NMR assay to achieve saturation.

In the context of tail acetylation, the hypothesis that acetylation affects tail accessibility has been tested previously by Stutzer et al., study, where they show that one tail mark can promote general accessibility for depositing other tail marks by specific enzymes. For the work in this study to go beyond the published work would require directly assessing the effect of nucleosomal tail acetylation (or other modifications) on BPTF binding either by NMR or other methods.

In addition, a more explicit comparison of the results from this study with the results from Stutzer et al., is needed throughout the text to provide the appropriate context for identifying and appreciating the new findings made in this study.

As it is unclear how long it would take to acquire the necessary data for increasing the impact of the work, the reviewers concluded that the current manuscript should be rejected, rather than be caught in what might be a considerable delay.

Reviewer #1:

The identification of new post-translational modifications (PTMS) on histone tails and reader domains that specifically recognize these modifications continues to grow at a rapid pace. To date most of the quantitative characterization of readers with histone PTMs has been carried out in the context of tail peptides. However, it is also known that in the context of a nucleosome the histone tails interact with nucleosomal DNA. Thus, understanding the structural and energetic consequences of these two competing interactions is essential to understand the full regulatory potential if histone tail modfications. Here the authors use NMR approaches to quantify the energetic effects of nucleosome context on the ability of a key histone reader, the BPTF PHD finger to bind the H3-K4me3 mark. They present data to suggest that binding of the BPTF PHD finger is reduced by ~ 100fold in the context of the nucleosome and that this inhibition can be explained by competition with DNA binding. Through the use of NMR, they further relate the quantitative effects to H3 tail dynamics and suggest that the nucleosomal H3 tail is in fast exchange between a nucleosomal DNA bound state and a free state. They also present molecular dynamics data to support the NMR results. Based on these experiments the authors propose a model in which interactions with the nucleosomal DNA reduce the accessibility of the nucleosomal H3 tail by ~100-fold compared to a free H3 tail peptide and that other PTMs such as acetylation, which would reduce DNA interactions, can increase H3 tail accessibility for binding by reader domains.

Overall the experiments are rigorously carried out with the key controls, including important controls for the effects of the MLA, and the conclusions are well supported. From the perspective of impact, this appears to be the first quantification of the effect that nucleosomal context has on the intrinsic out vs. in equilibrium of the H3 tail, and the magnitude the effect implies a large regulatory dynamic range that can be tuned by other PTMS as well proteins that bind nucleosomal DNA. The types of models opened up by such considerations are more sophisticated than current simple models of combinatorial PTM recognition. Therefore, this work is likely to substantially enrich discussion in the field about the mechanisms by which histone modifications regulate genome accessibility.

Reviewer #2:

This manuscript is a composite of replication of prior work by others and conceptually interesting, imperfectly executed (or insufficiently detailed in supporting information) so that strong new conclusions cannot be drawn. I wanted to like this paper, however I have serious reservations about many of the experiments that represent interesting departures from that which has already been done. Were many of these experiments cleaned up, this manuscript could represent a conceptually important advance in the understanding of how the restricted H3 tail dynamics that occur as a consequence, interactions with nucleosomal DNA can impact binding partners that engage tail-embedded marks. However, the existing data falls short of conclusively demonstrating that the effects observed are due to previously noted H3 tail-DNA interactions, rather than MLA differences coupled with a titration that gets nowhere near saturation so that quantification of affinity is not appropriate.

A clear comparison point for this work is the elegant study that also used NMR of labeled H3 to demonstrate the more limited dynamics in the H3 tail (Stutzer et al., 2016). Confirmation is an important function of science, but this seems too high profile a journal for this to be the basis of publication. The present manuscript confirms previously published work's conclusion that the H3 tail is adopts a heterogenous ensemble of conformations that are in close contact with DNA, and these columbic interactions between basic residues and DNA can be weakened by tail PTMs that neutralize or impart negative charge. The prior work also did a very similar H3 tail titration against DNA, and found much the same thing, although adding H3K4me3 and PHD finger is a new twist. In revision, the authors should emphasize new experiments rather than ones that are essentially repeating prior work-these latter experiments, though nice are more germane to supplementary display. The tail-less experiment with H3 peptide titration is an exciting and new experiment, and the MD simulations add a modicum of additional insight. Although the calculations of free energy seem a bit too far out.

My biggest concern is that the NMR titrations that do not saturate, or any quantitative affinity measurement that does not saturate cannot be used to compute accurate Kd values-one can only say that the binding is weaker, but not that it is "[…]nearly two orders of magnitude weaker than that measured for the MLA-peptide." Secondarily, I am not sure how the staging of NMR titrations in stoichiometric binding regime (well above computed Kd for peptides) can be used to compute accurate Kd-saturation or near saturation is reached when the peptide becomes equimolar with protein. My sense is that most NMR titrations provide values that are very different (orders of magnitude in some cases) than those measured by other quantitative equilibrium methods: FP, ITC, SPR, MST, etc. So, validation of the peptide and MLA affinities by an orthogonal more accurate means would greatly strengthen the conclusions, and would be great at the nucleosome level, although very challenging. Furthermore, as no characterization of the MLA histone is presented so the lower affinity could be due in the part or whole to incomplete or excessive alkylation. I also worry that peptide quantification by mass is not sufficiently accurate. These points are critical to the interpretation of weaker binding as a consequence tail engagement with DNA.

Reviewer #3:

The revised manuscript from Morrison and colleagues describes a series of NMR and MD simulation studies that indicate that interactions between the N-terminal tail of histone H3 and nucleosomal DNA are a critical factor in regulating the binding of PhD finger domain protein to H3 K4 trimethylation. This manuscript significantly adds to a recently emerging awareness that the N-terminal tails of the histones do not function independently as was assumed by the traditional models showing the tails extending out form the surface of a nucleosome. Rather, this manuscript (and a few others) shows that the N-terminal tails are dynamically associated with the core of the nucleosome and with the nucleosomal and linker DNA. The data in the manuscript is convincing and the conclusions drawn are reasonable and will be impactful. However, there are two issues with the manuscript. First, the data used to examine the effect of the MLA on binding of BPTF relative to trimethyllysine. The authors conclude that the MLA has a negative effect of BPTF binding. However, this seems to be based on an apples to oranges comparison. The binding of a peptide containing residues 1 to 10 of the H3 tail with K4 trimethylated is compared to binding a peptide containing H3 tail residues 1 to 44 with the MLA. While the peptide with the MLA shows lower affinity, it is possible that the change in affinity is due to potential structural changes in the peptide that exist in the 1 to 44 peptide but not the 1 to 10 peptide. The authors should compare peptides with the different modifications that have the same length. The second issue regards the final set of experiments examining the effect of acetylation on the interaction of the H3 tail with nucleosomal DNA. It would increase the relevance of these experiments if the simulations were tested by comparisons of binding BPTF to NCPs with H3 K4 trimethylation plus and minus acetylation. This may be a difficult experiment fot set up but without this direct test, the section of acetylation does not significantly add to the manuscript.

[Editors’ note: what now follows is the decision letter after the authors submitted for further consideration.]

Thank you for submitting your article "The conformation of the histone H3 tail inhibits association of the BPTF PHD finger with the nucleosome" for consideration by *eLife*. Your article has been reviewed by two peer reviewers, and the evaluation has been overseen by a Reviewing Editor and Jessica Tyler as the Senior Editor. The following individual involved in review of your submission has agreed to reveal his identity: Mark Parthun (Reviewer #3).

The reviewers have discussed the reviews with one another and the Reviewing Editor has drafted this decision to help you prepare a revised submission.

As with the first submission the reviewers think that quantifying the magnitude of inhibitory effects of the nucleosome on reader binding is of biological significance as it uncovers the dynamic range of tail accessibility that can be tuned by other nucleosome modifiers. In response to the first submission the reviewers had asked for two major additions: (i) Bona-fide nucleosome Kd values and (ii) the effects of PTMs on reader binding in the context of nucleosomes. The reviewers appreciate that your resubmission now has data on the effects of PTMs and think this new data adds to the significance of the work. However, the lack of saturation in the context of Kd values for nucleosomes, is still somewhat of a concern as is the emphasis on experiments that largely confirm previous findings by Stutzer et al., 2016. Below we outline how these, and related concerns could be addressed in a revised manuscript.

Essential revisions:

1) Given that saturation is not achieved it is essential that the Kd values be reported as limits and not as estimates based on assumptions about the ppm changes at saturation. This is because it is possible that the maximal ppm changes in the context of a nucleosome may differ from those in the context of a free peptide. In the revised manuscript, while the initial Results text is suitably softened in this context, in the section describing H3 tail charge mutant nucleosome NMR measurements (subsection “Modification and mutation of the H3 tail increase accessibility to the PHD finger”) approximate Kd values are reported. Further, in Figure 9, Kds are reported with a "=" rather than a "~", as well as associated uncertainty (which also cannot be accurately computed), and in Table 1. The Kd values should instead be reported as > the highest concentration of nucleosomes used. Given such a measure, from the data in Figure 9 it is still possible to say that the nucleosome inhibits reader accessibility to tail residues by > 20-fold.

2) Given that saturation is not achieved, one concern is whether the reader is still binding specifically to the H3Kc4me3 or whether it is binding the nucleosome non-specifically at multiple other locations. In this context, a control to show that binding to unmethylated nucleosomes does not give the same chemical shifts is needed.

3) The overall paper still has many measurements with H3 tails and DNA that are quite similar to the prior Stutzer paper that came up in the first round of review, and the reviewers acknowledge that the authors now more effectively highlight the prior work as useful points of comparison. However, the portion of the manuscript that is dedicated to confirming previous work should be further de-emphasized in the text and main figures, so that the data that is original-how these nucleosomal H3 tail properties impact binding by the BPTF PHD finger-stands out more readily. The reviewers feel there is limited value in presenting identical or substantially similar data for example, subsecgtion “The nucleosome inhibits association of the BPTF PHD finger with the H3 tail” and subsection “The H3 tail experiences distinct conformations in isolation or in the native context of the NCP”.

4) A recent publication (Gatchalian et al., https://www-ncbi-nlm-nih-gov.ucsf.idm.oclc.org/pubmed/29138400) describes a similar phenomenon in the context of paired PHD finger readers. Here they are able to measure Kds for nucleosomal H3 tails using fluorescence polarization (Figure 1F of this paper) and find a ~ 6-fold weaker affinity on the nucleosome vs. free peptide. This presents an interesting and likely informative contrast to the work under review. The authors should therefore compare the effects in their study with those of Gatchalian et al., and discuss models for why the magnitude of the effects may be different given that the reader domains in both cases bind a similar region of H3.

---

## [Author Response]

[Editors’ note: the author responses to the first round of peer review follow.]

Reviewer #2:This manuscript is a composite of replication of prior work by others and conceptually interesting, imperfectly executed (or insufficiently detailed in supporting information) so that strong new conclusions cannot be drawn. I wanted to like this paper, however I have serious reservations about many of the experiments that represent interesting departures from that which has already been done. Were many of these experiments cleaned up, this manuscript could represent a conceptually important advance in the understanding of how the restricted H3 tail dynamics that occur as a consequence, interactions with nucleosomal DNA can impact binding partners that engage tail-embedded marks. However, the existing data falls short of conclusively demonstrating that the effects observed are due to previously noted H3 tail-DNA interactions, rather than MLA differences coupled with a titration that gets nowhere near saturation so that quantification of affinity is not appropriate.A clear comparison point for this work is the elegant study that also used NMR of labeled H3 to demonstrate the more limited dynamics in the H3 tail (Stutzer et al., 2016). Confirmation is an important function of science, but this seems too high profile a journal for this to be the basis of publication. The present manuscript confirms previously published work's conclusion that the H3 tail is adopts a heterogenous ensemble of conformations that are in close contact with DNA, and these columbic interactions between basic residues and DNA can be weakened by tail PTMs that neutralize or impart negative charge. The prior work also did a very similar H3 tail titration against DNA, and found much the same thing, although adding H3K4me3 and PHD finger is a new twist. In revision, the authors should emphasize new experiments rather than ones that are essentially repeating prior work-these latter experiments, though nice are more germane to supplementary display. The tail-less experiment with H3 peptide titration is an exciting and new experiment, and the MD simulations add a modicum of additional insight. Although the calculations of free energy seem a bit too far out.My biggest concern is that the NMR titrations that do not saturate, or any quantitative affinity measurement that does not saturate cannot be used to compute accurate Kd values-one can only say that the binding is weaker, but not that it is "[…]nearly two orders of magnitude weaker than that measured for the MLA-peptide."

We agree that we cannot obtain an accurate Kd value from data that has not reached saturation. We modified the text to be abundantly clear that this is an approximate Kd value. And so as to not risk over-interpretation we have removed any quantitative comparison between the Kds for peptide and NCP, and simply note that the interaction with the nucleosome is substantially weaker than that observed for peptide.

Secondarily, I am not sure how the staging of NMR titrations in stoichiometric binding regime (well above computed Kd for peptides) can be used to compute accurate Kd-saturation or near saturation is reached when the peptide becomes equimolar with protein. My sense is that most NMR titrations provide values that are very different (orders of magnitude in some cases) than those measured by other quantitative equilibrium methods: FP, ITC, SPR, MST, etc. So, validation of the peptide and MLA affinities by an orthogonal more accurate means would greatly strengthen the conclusions, and would be great at the nucleosome level, although very challenging.

Indeed, the protein concentration for the NMR studies (50 μM) is above the computed Kd for the peptides, hence why we took into account ligand depletion in calculating the Kd value. However, we appreciate that we are nearing stoichiometric conditions for the peptides. We have remeasured the Kd for the H3KC4me3(1-44) histone peptide by biolayer interfermoetery (BLI). Notably, the binding to the H3KC4me3-NCP was not detectable by BLI, confirming the NMR results that binding to the nucleosome is substantially weaker. For the short peptide, we removed the computed NMR Kd, and did not pursue quantitative analysis by BLI, as a very nice report published during revision of this work thoroughly investigates the difference between true methyl-lysine and the MLA specifically for the BPTF PHD finger. We discuss this work in the revised text (see Chen, 2017).

Furthermore, as no characterization of the MLA histone is presented so the lower affinity could be due in the part or whole to incomplete or excessive alkylation.

We apologize for not previously including this critical data. We agree that it is absolutely essential to demonstrate that the histone proteins are not excessively alkylated or carbamylated. We have included mass spec data in the revised version to address this (see Figure 1—figure supplement 3 and Figure 9—figure supplement 2).

I also worry that peptide quantification by mass is not sufficiently accurate. These points are critical to the interpretation of weaker binding as a consequence tail engagement with DNA.

We agree that there could be inaccuracies introduced by mass measurement of peptides (which was carried out for the short peptide). However, the longer peptide that we make the bulk of our comparisons with includes a native tyrosine (Y41) allowing us to quantitate concentration spectrophotometrically. This is clarified in the revised Materials and methods section.

Reviewer #3:The revised manuscript from Morrison and colleagues describes a series of NMR and MD simulation studies that indicate that interactions between the N-terminal tail of histone H3 and nucleosomal DNA are a critical factor in regulating the binding of PhD finger domain protein to H3 K4 trimethylation. This manuscript significantly adds to a recently emerging awareness that the N-terminal tails of the histones do not function independently as was assumed by the traditional models showing the tails extending out form the surface of a nucleosome. Rather, this manuscript (and a few others) shows that the N-terminal tails are dynamically associated with the core of the nucleosome and with the nucleosomal and linker DNA. The data in the manuscript is convincing and the conclusions drawn are reasonable and will be impactful. However, there are two issues with the manuscript. First, the data used to examine the effect of the MLA on binding of BPTF relative to trimethyllysine. The authors conclude that the MLA has a negative effect of BPTF binding. However, this seems to be based on an apples to oranges comparison. The binding of a peptide containing residues 1 to 10 of the H3 tail with K4 trimethylated is compared to binding a peptide containing H3 tail residues 1 to 44 with the MLA. While the peptide with the MLA shows lower affinity, it is possible that the change in affinity is due to potential structural changes in the peptide that exist in the 1 to 44 peptide but not the 1 to 10 peptide. The authors should compare peptides with the different modifications that have the same length.

We agree that this is not a direct comparison, as pointed out in the text. While we were revising this paper, a manuscript was published that compares binding of the BPTF PHD finger to a histone peptide containing a true methylated lysine with one containing the MLA in great detail, including a structure (see Chen, 2017). Thus, we did not pursue this in our revised manuscript.

The second issue regards the final set of experiments examining the effect of acetylation on the interaction of the H3 tail with nucleosomal DNA. It would increase the relevance of these experiments if the simulations were tested by comparisons of binding BPTF to NCPs with H3 K4 trimethylation plus and minus acetylation. This may be a difficult experiment fot set up but without this direct test, the section of acetylation does not significantly add to the manuscript.

We agree and have now carried out experiments to address this. Specifically, we have tested the interaction of the PHD finger with a nucleosome containing four lysine-to-glutamine mutations in the H3 tail, which mimics acetylated lysine. As predicted, these led to a moderate increase in the ability of the PHD finger to bind. We also tested binding to a nucleosome containing three arginine-to-alanine mutations along the H3 tail, and one containing two phosphorylated serines. Similar to the glutamine construct, these led to a moderate increase in accessibility to PHD finger binding. Together this supports the model generated and the concept that it mediates cross-talk between residues along the H3 tail. This new data is summarized in Figure 9.

[Editors' note: the author responses to the re-review follow.]

Essential revisions:1) Given that saturation is not achieved it is essential that the Kd values be reported as limits and not as estimates based on assumptions about the ppm changes at saturation. This is because it is possible that the maximal ppm changes in the context of a nucleosome may differ from those in the context of a free peptide. In the revised manuscript, while the initial Results text is suitably softened in this context, in the section describing H3 tail charge mutant nucleosome NMR measurements (subsection “Modification and mutation of the H3 tail increase accessibility to the PHD finger”) approximate Kd values are reported. Further, in Figure 9, Kds are reported with a "=" rather than a "~", as well as associated uncertainty (which also cannot be accurately computed), and in Table 1. The Kd values should instead be reported as > the highest concentration of nucleosomes used. Given such a measure, from the data in Figure 9 it is still possible to say that the nucleosome inhibits reader accessibility to tail residues by > 20-fold.

We understand and appreciate the reviewer’s concern regarding the assumption of an identical bound chemical shift. As we are using a quadratic instead of a simple hyperbola to fit the data we are not comfortable reporting the lower limit in this manner, so we instead have removed all Kd approximations for the nucleosomes and simply state that the CSPs are consistent with low millimolar affinity for the H3KC4me3-NCP, and discuss trends in the CSPs for each of the additionally modified nucleosomes without any mention of Kd.

2) Given that saturation is not achieved, one concern is whether the reader is still binding specifically to the H3Kc4me3 or whether it is binding the nucleosome non-specifically at multiple other locations. In this context, a control to show that binding to unmethylated nucleosomes does not give the same chemical shifts is needed.

This is an excellent point. We have performed this control titration (see Figure 1—figure supplement 4). The titration confirms that PHD finger binding to the nucleosome is dependent on methylation.

3) The overall paper still has many measurements with H3 tails and DNA that are quite similar to the prior Stutzer paper that came up in the first round of review, and the reviewers acknowledge that the authors now more effectively highlight the prior work as useful points of comparison. However, the portion of the manuscript that is dedicated to confirming previous work should be further de-emphasized in the text and main figures, so that the data that is original-how these nucleosomal H3 tail properties impact binding by the BPTF PHD finger-stands out more readily. The reviewers feel there is limited value in presenting identical or substantially similar data for example, subsecgtion “The nucleosome inhibits association of the BPTF PHD finger with the H3 tail” and subsection “The H3 tail experiences distinct conformations in isolation or in the native context of the NCP”.

We agree. Though we wanted to be sure that the comparisons being made were clear, the reviewers’ feedback confirms that it is, and thus we have removed previous text in the Results section as well as substantial text in subsection “The H3 tail experiences distinct conformations in isolation or in the native context of the NCP.” to de-emphasize these points. We note that as the data shown here is the first NMR data on human histones with only 147bp of DNA, it is important to emphasize what about these spectra are the same as seen for previous constructs. Therefore, some of the comparison remains, though it is not belabored. In addition, the data in Figure 4 regarding binding of the H3 tail to DNA has been moved to supplemental and the associated text substantially shortened.

4) A recent publication (Gatchalian et al., https://www-ncbi-nlm-nih-gov.ucsf.idm.oclc.org/pubmed/29138400) describes a similar phenomenon in the context of paired PHD finger readers. Here they are able to measure Kds for nucleosomal H3 tails using fluorescence polarization (Figure 1F of this paper) and find a ~ 6-fold weaker affinity on the nucleosome vs. free peptide. This presents an interesting and likely informative contrast to the work under review. The authors should therefore compare the effects in their study with those of Gatchalian et al. and discuss models for why the magnitude of the effects may be different given that the reader domains in both cases bind a similar region of H3.

We agree that this is an important comparison and have added this to the Discussion section regarding other reader/NCP studies.